# DDX1 vesicles control calcium-dependent mitochondrial activity in mouse embryos

Yixiong Wang[1], Lubna Yasmin[1], Lei Li[1], Pinzhang Gao[1], Xia Xu[1], Xuejun Sun[1] & Roseline Godbout [1✉]

The DEAD box protein DDX1, previously associated with 3'-end RNA processing and DNA repair, forms large aggregates in the cytoplasm of early mouse embryos. *Ddx1* knockout causes stalling of embryos at the 2-4 cell stages. Here, we identify a DDX1-containing membrane-bound calcium-containing organelle with a nucleic acid core. We show that aggregates of these organelles form ring-like structures in early-stage embryos which we have named Membrane Associated RNA-containing Vesicles. We present evidence that DDX1 is required for the formation of Membrane Associated RNA-containing Vesicles which in turn regulate the spatial distribution of calcium in embryos. We find that *Ddx1* knockout in early embryos disrupts calcium distribution, and increases mitochondria membrane potential, mitochondrial activity, and reactive oxygen species. Sequencing analysis of embryos from *Ddx1* heterozygote crosses reveals downregulation of a subset of RNAs involved in developmental and mitochondrial processes in the embryos with low *Ddx1* RNA. We propose a role for Membrane Associated RNA-containing Vesicles in calcium-controlled mitochondrial functions that are essential for embryonic development.

[1] Department of Oncology, Cross Cancer Institute, University of Alberta, Edmonton, AB T6G 1Z2, Canada. ✉email: rgodbout@ualberta.ca

There are two waves of zygotic genome activation during early-stage development: a minor wave at the 1-cell stage and a major wave at the 2-cell stage[1]. The majority of the zygotic genome is activated after these two waves of zygotic genome activation[2]. Tightly orchestrated regulation of maternal factors including RNAs, proteins and mitochondria, is critical for early embryonic development[2,3]. Most embryonic arrests at the 2-cell stage are related to maternal factors. However, a few zygotic genes have also been associated with early embryonic lethality including *Pp2cβ*[4], *Ddx20*[5], *Pfn1*[6], *Ercc2*[7] and *Ddx1*[8]. DDX1 is an RNA binding/unwinding protein that is primarily found in the nucleus of somatic cells. However, in early-stage mouse embryos, DDX1 localizes to the cytoplasm where it forms large aggregates and co-compartmentalizes with the stress granule markers TIA-1 and TIAR[8]. *Ddx1* knockout in *Drosophila melanogaster* suggests a role for DDX1 in RNA splicing and mitochondria function[9].

Deregulation of RNA and mitochondrial function, both of which are controlled by calcium distribution[10–12], are major causes of early embryonic lethality. In oocytes, mitochondrial integrity has been shown to be key to fertilization and embryonic development. It is therefore important to identify potential causes and factors contributing to mitochondrial dysfunction[13–15]. However, despite being an essential player in the control of mitochondrial activity during embryonic development[16,17], the exact function and location of calcium remains poorly characterized.

Here, we show that calcium is located in membrane-bound organelles loosely clustered into ring-like structures that we have named MARVs (Membrane Associated RNA-containing Vesicles). Each membrane-bound vesicle has a core of RNA and is enriched in the DEAD Box 1 (DDX1) protein. *Ddx1* knockout in mice reduces MARV formation and causes nuclear and mitochondrial fragmentation in 2-cell embryos. We propose that MARVs represent a heretofore missing link in our understanding of calcium-dependent activation of early events in mammalian embryonic development particularly as related to mitochondria activity.

## Results

### DDX1 aggregates consist of large membrane-bound vesicles.
We used STimulated Emission Depletion (STED) super-resolution microscopy combined with Atto 550-conjugated anti-DDX1 antibody to resolve the structure of DDX1 aggregates previously identified by confocal microscopy in 2-cell mouse embryos[8]. We discovered that each DDX1 aggregate consists of a ring-like cluster (Fig. 1a), with these clusters gradually becoming larger, denser, and less ring-like with embryonic development (Supplementary Fig. 1a). Interestingly, DDX1 aggregates are primarily found in the subplasmalemmal cytoplasm, spatially located immediately next to the plasma membrane (Supplementary Fig. 1b). It has previously been reported that mitochondria with high membrane potential ($\Delta\Psi_m$), generally associated with high levels of mitochondrial activity, also localize to the subplasmalemmal cytoplasm[18]. To further investigate the distribution of mitochondria with high $\Delta\Psi_m$ in 2-cell embryos, we used a positively charged mitochondrial potential indicator ratiometric dye, JC-1. Mitochondrial $\Delta\Psi_m$ is measured by comparing the fluorescence signal at 590 nm (JC-1 in J-aggregates in active mitochondria) versus 529 nm (JC-1 in monomeric form in inactive mitochondria). As previously published[18], mitochondria with high $\Delta\Psi_m$ (magenta) are located at the subplasmalemmal cytoplasm whereas the majority of mitochondria with low $\Delta\Psi_m$ (green) are located at the inner cytoplasm (Supplementary Fig. 1c). Thus, both mitochondria with high $\Delta\Psi_m$ and DDX1 aggregates are found in similar locations within 2-cell embryos.

To examine the ultrastructure of DDX1 aggregates, we used transmission electron microscopy (TEM) with DDX1-immunogold-labelled 2-cell embryos. In agreement with our STED data, DDX1-containing aggregates localized to the subplasmalemmal cytoplasm, with many aggregates located next to mitochondria (Fig. 1b). Unexpectedly, we found that each DDX1 aggregate[8] consists of several large previously uncharacterized membrane-bound vesicles (~200−400 nm) loosely structured in a ring (Fig. 1b, c). These ring-like structures are generally located $1.46 \pm 0.55\,\mu m$ ($n = 24$) from the plasma membrane.

Next, we used TEM to determine when membrane-bound DDX1 vesicles appear during development. Previous confocal microscopy analysis showed abundant small DDX1 granules/aggregates in MII oocytes[8]. TEM showed absence of membrane-bound DDX1 vesicles in MII oocytes (Supplementary Fig. 2a). Membrane-bound vesicles first appeared in 1-cell embryos (Supplementary Fig. 2a). In 1-cell embryos (E0.5), $40.0 \pm 18.5\%$ of DDX1-immunogold particles were detected in membrane-bound DDX1 vesicles ($n = 20$ images), compared to $83.7 \pm 3.6\%$ ($n = 20$), $83.2 \pm 5.6\%$ ($n = 20$) and $85.6 \pm 3.2\%$ ($n = 19$) in E1.5, E2.5 and E3.5 embryos, respectively. With regards to E3.5 embryos, DDX1 vesicles were observed in both trophectoderm and inner cell mass cells, as previously reported[8] (Supplementary Fig. 2b). By late E4.5 blastocyst stage, membrane-bound DDX1 vesicles were no longer observed (Supplementary Fig. 2a). Our data indicate an abundance of DDX1 in vesicles compared to the rest of the embryo, suggesting that DDX1 can be used as a marker for these vesicles. Thus, a single DDX1 aggregate observed by confocal microscopy consists of multiple membrane-bound DDX1 vesicles that first appear in 1-cell embryos.

DDX1 is a member of the DEAD box family of RNA binding/unwinding/remodelling proteins[19]. We have previously shown that digestion of 2-cell embryos with RNase A causes disaggregation of DDX1[8]. To further investigate the relationship between RNA and DDX1 vesicles, we used electron microscopy and $OsO_4$/uranyl acetate staining which stabilizes nucleic acids and increases electron density[20]. We observed intense staining in a distinct patch within each vesicle (Fig. 1c). We next used energy-filtered transmission electron microscopy (EFTEM) for elemental mapping of nitrogen (indicative of proteins) and phosphate (indicative of nucleic acids) in 2-cell embryos[21]. Similar to what was observed with $OsO_4$/uranyl acetate staining, phosphorus was enriched in a clear compartment within each vesicle of each DDX1 aggregate (Fig. 1d). Combined with our previous RNase A data, these results indicate that RNA occupies a distinct compartment within each vesicle, suggesting a requirement for compaction, organization and/or localization of RNAs at these sites (Fig. 1e). As these compartmentalized RNA-containing membrane–bound vesicles have not been described previously, we named the ring-like aggregates of DDX1 vesicles 'Membrane Associated RNA-containing Vesicles' (MARVs).

### MARVs do not co-compartmentalize with ribosomes.
Embryos undergoing rapid cell division require a continuous supply of energy. The total number of mitochondria per embryo remains unchanged from the 1-cell to early blastocyst stages, with numbers of mitochondria per cell halved at each cell division[22,23], suggesting a need for increased mitochondrial function to compensate for their reduced numbers. We therefore asked whether MARVs might be associated with translation of proteins required for mitochondrial function and/or early embryonic development. First, we co-immunostained 2-cell embryos with anti-DDX1 antibody and antibodies to cytoplasmic ribosomal protein (RPS6), or mitoribosomal proteins MRP-L42, MRP-L44 or

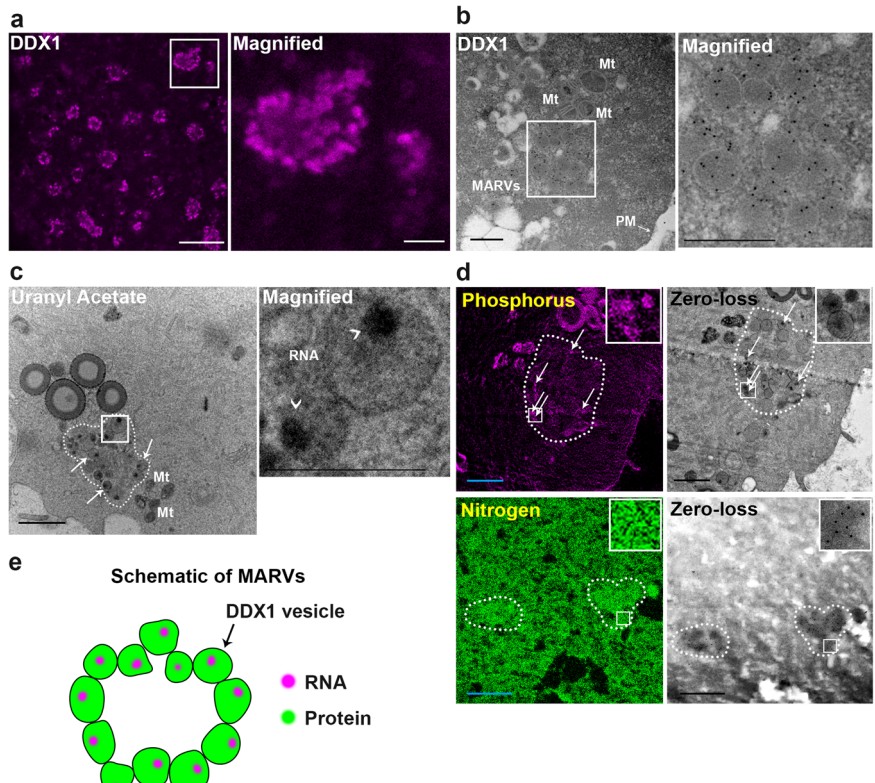

**Fig. 1 DDX1 aggregates form ring-like structures with regionalized RNAs in 2-cell embryos. a** STED image of DDX1 staining (magenta) in 2-cell embryos. The white square is magnified in the panel on the right. Scale bar = 5 μm. Magnified image scale bar = 1 μm. Four wild-type crosses were used for each stage (~8 embryos from each cross). Similar results were obtained for all embryos. **b** TEM image of immunogold-labelled DDX1 in a 2-cell embryo. An aggregate of DDX1 vesicles (MARV) is shown in the square magnified on the right. Scale bars = 0.5 μm. Four embryos from 2 wild-type crosses were independently collected with similar results obtained for all embryos. **c** TEM image showing 2-cell embryo stained with OsO$_4$/uranyl acetate. Arrows point to a subset of DDX1 vesicles within MARVs. Six embryos from 3 wild-type crosses were independently collected with similar results obtained for all embryos. Regions containing RNA are indicated by arrowheads. Scale bar = 1 μm. Magnified image scale bar = 0.5 μm. Mt: mitochondria. **d** Phosphorus (magenta) and nitrogen (green) EFTEM mapping in 2-cell embryos. Scale bars = 1 μm. Zero-loss images are also shown. Four embryos from 2 wild-type crosses were independently collected where similar results obtained for all embryos. A MARV is circled with the dotted line in (**c**), (**d**). **e** Graphical depiction of MARVs.

MRP-S27. DDX1 did not co-compartmentalize with any of these ribosomal proteins (Supplementary Fig. 3a, b). Next, we treated 1-cell embryos in culture with the global inhibitor of protein synthesis, cycloheximide, for 8 h. STED microscopy revealed decreased DDX1 aggregates (Supplementary Fig. 4), indicating that clustering of these vesicles is dependent on active translation. Conversely, treatment with chloramphenicol, an inhibitor of mitochondrial translation, for 8 h did not affect the DDX1 aggregates in the embryos, suggesting a link between MARVs and cytoplasmic, rather than mitochondrial, protein translation (Supplementary Fig. 4).

**Ca$^{2+}$ signal co-compartmentalizes with DDX1.** Ca$^{2+}$ signalling triggers the events required for embryonic development after fertilization including mitochondrial function[10,11,13,24,25] and regulation of mRNA translation[26]. Along with the endoplasmic reticulum, mitochondria control Ca$^{2+}$ release shortly after fertilization[26]. We used the Ca$^{2+}$ indicator Fluo-4 AM to study the distribution of Ca$^{2+}$ in 2-cell embryos in relation to that of mitochondria and DDX1 aggregates. As previously reported[16], we found that Ca$^{2+}$ was particularly abundant in subplasmalemmal cytoplasm with a similar distribution pattern to that of DDX1 aggregates (Fig. 2a). However, upon co-staining 2-cell embryos with Fluo-4 AM and MitoTracker Deep Red[27], we found no co-compartmentalization of Ca$^{2+}$ with mitochondria located at the

subplasmalemmal cytoplasm (Fig. 2b, e). As predicted, co-compartmentalization was observed when 2-cell embryos were co-stained with anti-DDX1 antibody and Fluo-4 AM (Fig. 2c, e). Furthermore, Fluo-4 AM did not co-compartmentalize with ER-Tracker in 2-cell embryos[28,29] (Fig. 2d, e), indicating that Ca$^{2+}$/DDX1-containing vesicles are unlikely to be cortical ER, a reported location of Ca$^{2+}$ in oocytes[30]. Our combined results are thus in line with Ca$^{2+}$/DDX1-containing vesicles serving as reservoirs for rapid mobilization of Ca$^{2+}$ and RNAs required for early embryonic development. In further agreement with our thinking, treatment of 2-cell embryos with the protonophore FCCP which abolishes mitochondrial membrane potential, disrupted Ca$^{2+}$ distribution as previously shown by others[16,17], indicating that these vesicles serve as Ca$^{2+}$ reservoirs associated with mitochondrial function. Compared to ER which is the major Ca$^{2+}$ reservoir in somatic cells, MARVs with their aggregates of membrane-bound vesicles may allow more precise control of Ca$^{2+}$-dependent mitochondrial functions.

**Ca$^{2+}$ affects the size of MARVs.** The above data indicate that Ca$^{2+}$ localizes to MARVs in 2-cell embryos. To further examine the link between MARVs and Ca$^{2+}$, we disrupted Ca$^{2+}$ distribution by treating 2-cell embryos with FCCP for 2 h. We observed a small but statistically significant increase in the number of small DDX1 aggregates (<0.5 μm$^3$) but not those

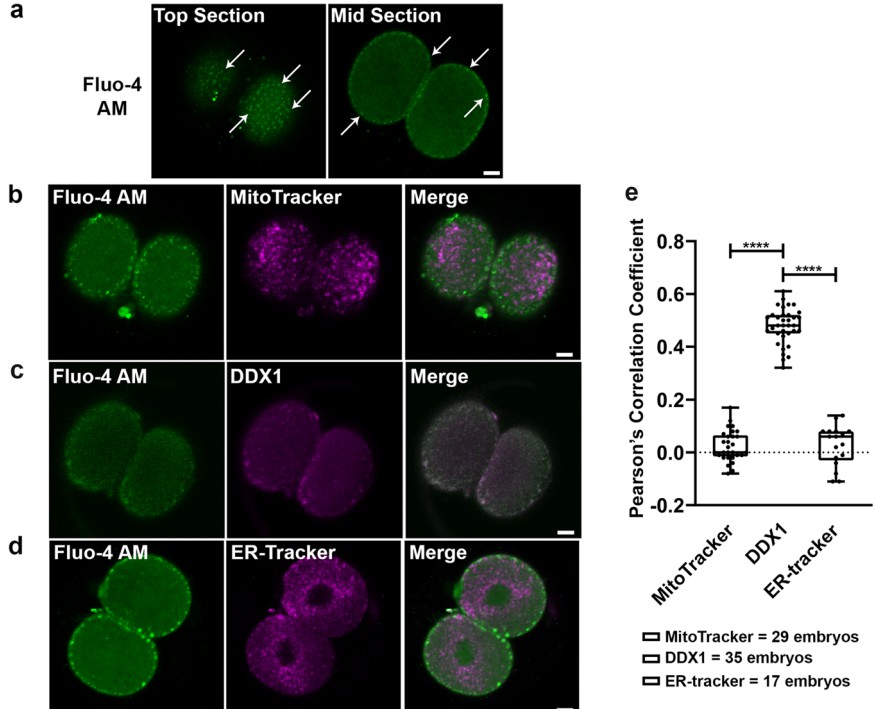

**Fig. 2 Co-compartmentalization of Ca²⁺ and DDX1 in MARVs but not ER or mitochondria. a** Ca²⁺ microdomains in 2-cell embryos show a similar distribution to that of DDX1. Arrows point to a subset of Ca²⁺ microdomains. **b** Ca²⁺ microdomains (Fluo-4 AM, green) do not co-compartmentalize with MitoTracker Deep Red (magenta). **c** Ca²⁺ microdomains (Fluo-4 AM, green) co-compartmentalizes with DDX1 (magenta) in 2-cell embryos. **d** Ca²⁺ microdomains (Fluo-4 AM, green) do not co-compartmentalize with ER-Tracker Red (magenta). **e** Statistical analysis of (**b**) Fluo-4 AM/Mitotracker, (**c**) Fluo-4 AM/DDX1 and (**d**) Fluo-4 AM/ER-Tracker, with 2-cell embryos obtained from 3, 4 and 2 pairs, respectively, of wild-type natural matings. Pearson's correlation coefficients were calculated by ImageJ and plotted with Prism. Statistical analysis of single plane confocal images of each embryo (middle sections) was performed with one-way ANOVA and Tukey multiple comparison test. ****indicates $p < 0.0001$. Center line, median; box limits, 25th and 75th percentiles; whiskers, minimum to maximum. Error bars represent standard deviation. Scale bars = 10 μm.

>5 μm³ suggesting a limited impact on the formation/aggregation of DDX1 vesicles which may be due to the relatively short FCCP treatment time (Fig. 3a, b). As other researchers have shown that 2-cell embryos cultured in Ca²⁺-free media can develop to the 8-cell stage with failed compaction and blastocyst formation, we cultured 2-cell embryos in Ca²⁺-free media for 24 h (to the 4-cell stage). We then examined Ca²⁺ distribution and the size of DDX1 aggregates. Although no change in Ca²⁺ distribution was noted in 4-cell embryos, the intensity of Ca²⁺ microdomains was decreased by an average of ~40% compared to embryos cultured under regular Ca²⁺ conditions (Supplementary Fig. 5a, b). Moreover, in addition to an increase in small DDX1 aggregates (<0.5 μm³) (Fig. 3c, d), the average number of DDX1 aggregates increased by ~75%, whereas the overall size of DDX1 aggregates decreased by an average of ~34% (Fig. 3e) in embryos cultured in Ca²⁺ free media compared to regular Ca²⁺ containing media. We next stained 2-cell embryos cultured in Ca²⁺ free media for 24 h (to the 4-cell stage) with JC-1 to see whether they were still metabolically active. Although there was an average of ~18.4% reduction in mitochondrial membrane potential (J-aggregate/monomer) compared to control embryos, all embryos remained metabolically active (Supplementary Fig. 5c, d) indicating that they are still alive and can undergo development. Therefore, the effect on DDX1 aggregates observed under Ca²⁺ free culture conditions is unlikely to be due to non-viability of the embryos. In agreement with previous publications[31–33], embryos cultured in Ca²⁺ free media failed to compact after 48 h culture (Supplementary Fig. 5f). These data indicate that the size and numbers of MARVs are affected by Ca²⁺ distribution and mitochondrial

membrane potential. However, these data do not eliminate the possibility that mitochondrial metabolic signalling plays a role in MARVs and Ca²⁺ localization. In order to address this possibility, we treated 2-cell embryos with EGTA-AM, a Ca²⁺ chelator, for 3 h. Similar to Ca²⁺ free culture, we observed an overall decrease in Ca²⁺ intensity when embryos were stained with Fluo-4 AM, with no obvious changes in Ca²⁺ distribution (Fig. 4a, b). We observed a significant increase in the number of small DDX1 aggregates (<0.5 μm³) and a decrease in the number of DDX1 aggregates >5 μm³ (Fig. 4c, d) suggesting a direct role for Ca²⁺ distribution/levels in DDX1 aggregates.

**Ca²⁺ relocates to mitochondria upon saponin treatment.** To further investigate the relationship between Ca²⁺ distribution and MARVs, we examined the Ca²⁺ distribution upon its release from membrane-bound structures. For these experiments, we used saponin, a mild detergent used for live cell permeabilization[8]. As saponin permeabilizes plasma membrane as well as organelle membranes, this treatment will result in the release of Ca²⁺ from all membrane-bound organelles including MARVs[34]. Staining of saponin-permeabilized embryos with Fluo-4 AM showed disruption of Ca²⁺ subplasmalemmal distribution, with Ca²⁺ moving to the inner cytoplasm (Fig. 4e). Based on immunogold-labeling TEM, $78.5 \pm 8.5\%$ ($n = 15$) of DDX1 remained in MARVs after saponin-permeabilization of embryos (Fig. 4f). Interestingly, upon co-staining with Fluo-4 AM and MitoTracker Deep Red after saponin treatment, we found co-compartmentalization of Fluo-4 AM and MitoTracker Deep Red, suggesting that Ca²⁺ had migrated to inner cytoplasm

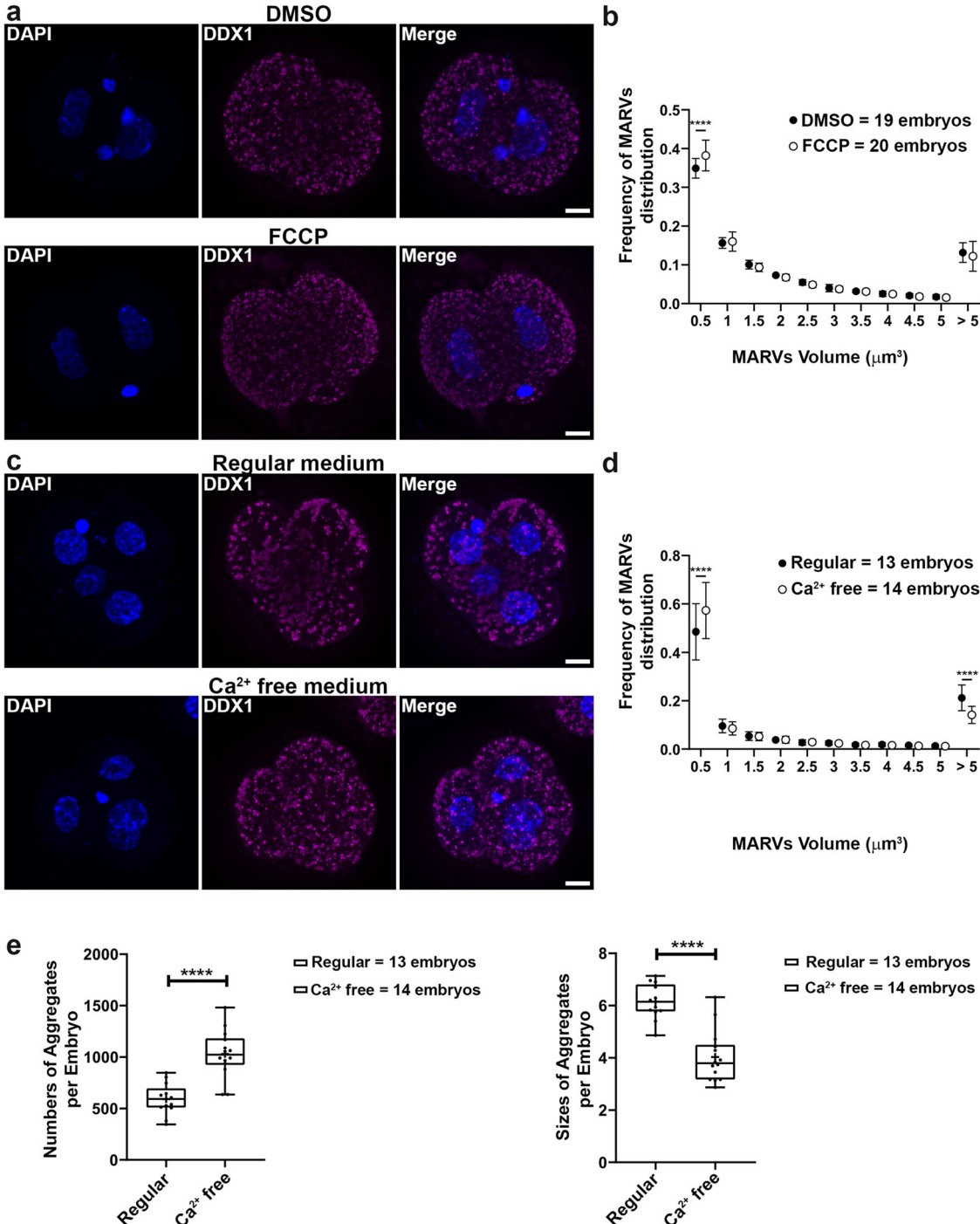

**Fig. 3 Sizes of MARVs are linked to mitochondrial potential and Ca$^{2+}$ distribution/supply.** Two-cell embryos were collected from natural matings. **a** Embryos were treated with 5 μM FCCP for 2 h. DDX1 aggregates showed reduced sizes in FCCP-treated embryos compared to DMSO control (DDX1, magenta; DAPI, blue). **b** Statistical analysis of (**a**) with data from 4 pairs of wild-type natural matings. Data were plotted with Prism. Statistical analysis was performed with two-sided multiple *t*-test using the Holm-Sidak method, with alpha = 0.05. ****indicates $p < 0.0001$ (with the exact $p < 1E-06$). Each circle represents the mean value, and the error bars represent standard deviation. Solid circles represent DMSO-treated control; empty circles represent FCCP-treated. **c** Embryos cultured in Ca$^{2+}$ free medium for 24 h show reduced sizes of DDX1 aggregates compared to embryos cultured in regular Ca$^{2+}$ containing medium. **d** Size distribution analysis of (**c**) with data from 3 pairs of wild-type natural matings. Data were plotted with Prism. Statistical analysis was performed with two-sided multiple *t*-test the Holm-Sidak method, with alpha = 0.05. ****indicates $p < 0.0001$ (with the exact $p < 1E-06$ for area $<0.5\ \mu m^3$ and $p = 4.5E-05$ for area $>5\ \mu m^3$). Each circle represents the mean value, and the error bars represent standard deviation. Solid circles represent regular medium; empty circles represent Ca$^{2+}$-free medium. **e** Average aggregate numbers and sizes analysis of (**c**). Data were plotted with Prism. Statistical analysis was performed with two-sided Students' *t*-test. ****indicates $p < 0.0001$ (with the exact $p = 4.26E-06$ for left panel and $p = 8.29E-07$ for right panel). Center line, median; box limits, 25th and 75th percentiles; whiskers, minimum to maximum. Error bars represent standard deviation. Maximum intensity projections of the Z-stack images of each stage are shown in (**a**), (**c**). Z-stacks were used for all statistical analyses in this figure. Scale bars = 10 μm.

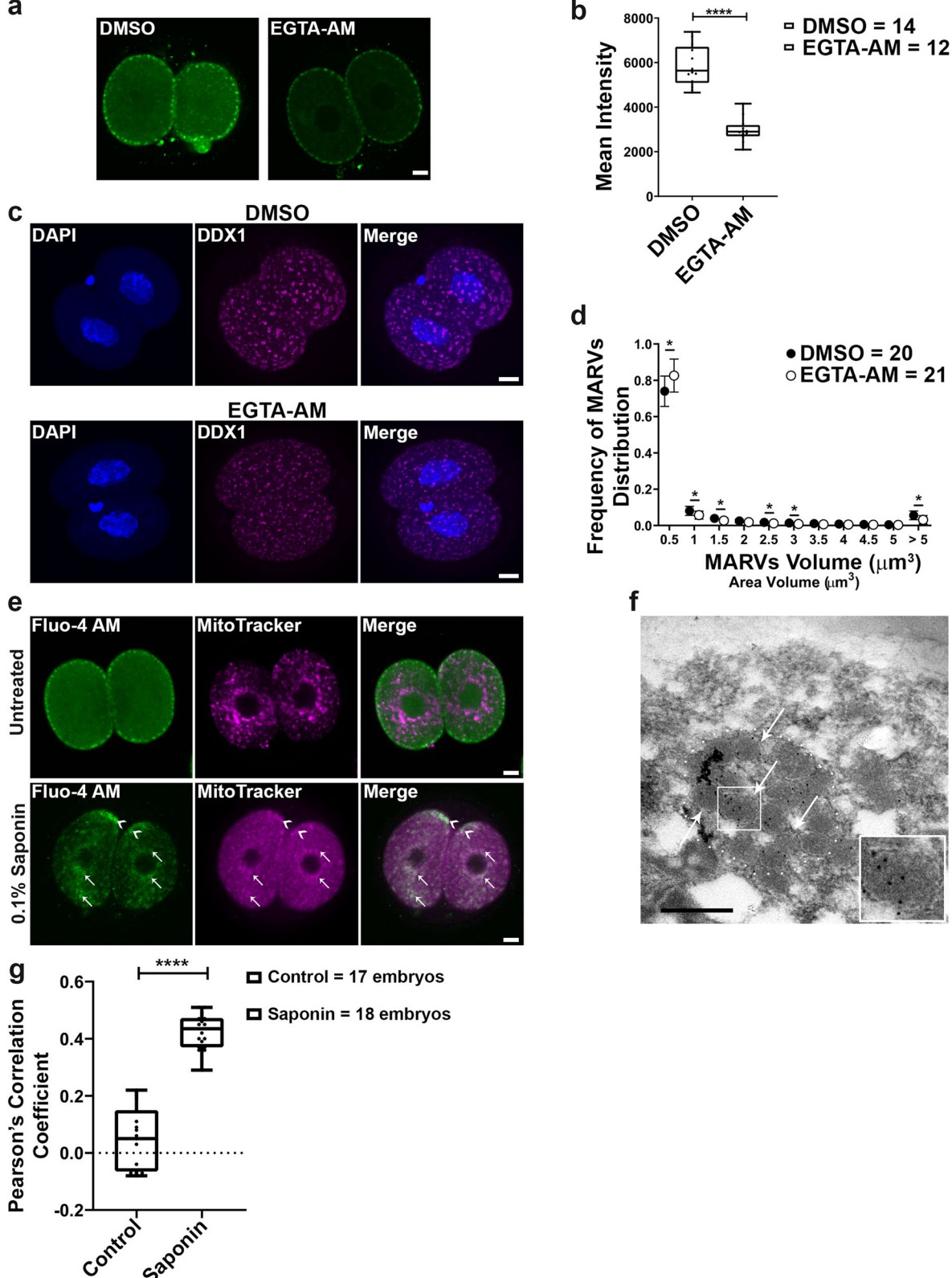

mitochondria upon permeabilization of DDX1/Ca$^{2+}$-containing vesicles (Fig. 4e, g). Even though saponin-permeabilization also permeabilizes the plasma membrane which will result in the leakage of Ca$^{2+}$ to the outside of the embryos, relocation of the remaining Ca$^{2+}$ inside the embryos to inner mitochondria points to a possible role for MARVs in regulating the spatial distribution of Ca$^{2+}$ in 2-cell embryos.

**$Ddx1^{-/-}$ embryos have high levels of mitochondrial activity.** We next cultured 1-cell embryos from $Ddx1^{+/-}$ intercrosses to address the effect of $Ddx1$ knockout on Ca$^{2+}$ distribution and embryonic development. Cultured 1-cell embryos from $Ddx1^{+/+}$ crosses were used for comparison. Ideally, the embryos from $Ddx1^{+/-}$ intercrosses should be imaged at the 2- to 4-cell stages and then cultured so that they could later be categorised into

**Fig. 4 Ca$^{2+}$ chelator EGTA affects the size of MARVs whereas saponin treatment causes redistribution of Ca$^{2+}$ to mitochondria.** Two-cell embryos were collected from natural matings. **a** Embryos were treated with 100 μM EGTA-AM or equivalent DMSO for 3 h. Ca$^{2+}$ microdomains (green) show similar but weaker patterns in EGTA-AM treated embryos compared to DMSO control. Scale bars = 10 μm. **b** Statistical analysis of (**a**) with data from 3 pairs of wild-type natural matings. Ca$^{2+}$ intensities were plotted with Prism. Statistical analysis was performed with two-sided Students' $t$-test. ****indicates $p < 0.0001$ (with the exact $p = 7.91E-10$). Center line, median; box limits, 25th and 75th percentiles; whiskers, minimum to maximum. Error bars represent standard deviation. **c** EGTA-AM treated embryos show reduced size of DDX1 aggregates compared to DMSO control (DDX1, magenta; DAPI, blue). Maximum intensity projections of the Z-stack images are shown. Scale bars = 10 μm. **d** Size distribution analysis of (**c**) with data from 4 pairs of wild-type natural matings. Data were plotted with Prism. Statistical analysis was performed with two-sided multiple $t$-test using the Holm-Sidak method, with alpha = 0.05. *indicates $p < 0.05$. The exact $p$ values are 0.024 for volumes <0.5 μm$^3$, 0.023 for volumes <1 μm$^3$, 0.046 for volumes <1.5 μm$^3$, 0.026 for volumes <2.5 μm$^3$, 0.021 for volumes <3 μm$^3$ and 0.021 for volumes >5 μm$^3$. Each circle represents the mean value, and the error bars represent standard deviation. Solid circles represent DMSO-treated controls; empty circles represent EGTA-treated embryos. **e** Co-staining of Ca$^{2+}$ (Fluo-4 AM; green) and mitochondria (MitoTracker, magenta) with or without saponin treatment. Arrows and arrowheads point to Fluo-4 AM and MitoTracker co-compartmentalization at the inner and subplasmalemmal cytoplasm, respectively. Scale bars = 10 μm. Images were obtained with a Zeiss LSM710 confocal microscope. **f** TEM shows that DDX1 remains in MARVs after treatment with the membrane permeabilizing glycoside, saponin. Arrows point to DDX1-containing vesicles. A MARV aggregate of DDX1 vesicles is outlined by the dots. Four embryos from 2 wild-type crosses were independently collected with similar results obtained for all embryos. Scale bar = 0.5 μm. **g** Statistical analysis of (**e**) with data collected from 2-cell embryos obtained from three pairs of wild-type natural matings (single plane mid-sections for each embryo). Pearson's correlation coefficients were calculated by ImageJ and plotted with Prism. Statistical analysis was performed with two-sided Students' $t$-test. ****indicates $p < 0.0001$ (with the exact $p = 3.39E-14$). Center line, median; box limits, 25th and 75th percentiles; whiskers, minimum to maximum. Error bars represent standard deviation.

stalled and normally developing embryos. However, due to the cytotoxicity of the dyes used for our experiments (MitoTracker Deep Red, MitoSOX, JC-1 and BioTracker ATP-red), and the phototoxicity of the laser scanning microscopy, we failed to obtain viable embryos after imaging 2- to 4-cell embryos in culture. We therefore had to first identify the stalled embryos after 72 h in culture before proceeding with these experiments. Analysis of 26 pairs of $Ddx1^{+/-}$ intercrosses revealed the expected ~28% (52/186) stalled embryos (presumed $Ddx1^{-/-}$) at the 2- to 4-cell stages after 72 h[8]. Please note that although $Ddx1^{-/-}$ embryos can be genotyped if taken directly from mice, we were not able to genotype stalled embryos, presumably because of the deterioration/modification in the DNA of stalled embryos. In our first set of experiments, we carried out single channel staining and quantification on stalled 2-cell embryos using Fluo-4 AM (5 embryos), JC-1 (13 embryos), and MitoSOX (7 embryos). We found little evidence of subplasmalemmal cytoplasmic Ca$^{2+}$ microdomains in stalled 2-cell embryos (72 h in culture) compared to wild-type 2-cell embryos (24 h in culture) (Fig. 5a). To ensure that the lack of subplasmalemmal cytoplasmic Ca$^{2+}$ microdomains was not the result of culturing embryos for 72 h, we also looked at cultured wild-type embryos at later developmental stages (4-cell, 16-cell and blastocyst, the latter cultured for 96 h). Similar to our previous results obtained upon DDX1 staining of blastocysts[8] we observed no clear difference in the Ca$^{2+}$ staining of ICM cells and trophectoderm cells. All wild-type embryos showed the expected immunostaining patterns based on in vivo DDX1 immunostaining patterns (Supplementary Fig. 1a). However, staining with JC-1 revealed higher mitochondrial membrane potential in stalled embryos compared to wild-type 2-cell embryos (Fig. 5b, c), with stalled embryos showing both mitochondria and nuclei fragmentation (Fig. 5d, e, f). Using MitoSOX, we also found increased levels of mtROS in stalled embryos compared to wild-type 2-cell embryos (Fig. 5g, h). As both JC-1 and MitoSOX staining suggest an increase in mitochondrial activity in stalled embryos, we next stained stalled embryos (15 embryos) with the ATP indicator dye, BioTracker ATP-red. In keeping with our JC-1 and MitoSOX results, increased ATP levels were observed in stalled embryos compared to the wild-type 2-cell embryos (Fig. 6a, b).

Single channel staining is optimal for generating quantification data as there is no channel-to-channel interference. However, to better document the relationship between mtROS, mitochondrial fragmentation and nuclear fragmentation, we also carried out

triple staining with Hoechst 33342 (DNA), MitoSOX (mtROS) and MitoTracker Deep Red (mitochondria) on 12 stalled embryos and 12 wild-type 2-cell embryos (Fig. 6c). Using the following three parameters (nucleus number/cell, MitoSOX intensity and percentage of mitochondria with size <0.2 μm$^2$) (Fig. 6d), we used weighted average index construction to generate an overall fatality index for stalled embryos versus wild-type embryo. The fatality index of stalled embryos was much higher than that of wild-type embryos (Fig. 6e).

**$Ddx1^{-/-}$ embryos show altered gene expression levels.** We have previously shown that DDX1 binds to a subset of RNAs that are essential for embryonic development[8]. To further investigate the effect $Ddx1$ knockout on embryonic RNA levels, we performed single embryo RNA sequencing on eighteen 2-cell embryos collected from 2 pairs of $Ddx1$ heterozygous crosses. Using normalized $Ddx1$ transcripts per million (TPM), we identified 2 presumed $Ddx1^{-/-}$ embryos, and 4 presumed wild-type embryos, with embryos with $Ddx1$ TPM levels ≤0.6 defined as low $Ddx1$ (or presumed $Ddx1^{-/-}$) and embryos with $Ddx1$ TPM levels >3 defined as high $Ddx1$ (or presumed wild-type). By comparing the sequencing data from the two low $Ddx1$ embryos with that of the four high $Ddx1$ embryos, we identified ~300 genes that were downregulated, and ~90 genes that were upregulated (FDR or false discovery rate <0.1), in low $Ddx1$ embryos compared to high $Ddx1$ embryos. Some of these genes play roles in embryonic development as well as mitochondrial function (Supplementary Dataset 1). A number of the identified RNAs were validated by RT-qPCR using independent biological samples [3 embryos with high levels of $Ddx1$ (or presumed wild-type) and 3 embryos with low levels of $Ddx1$ (or presumed $Ddx1^{-/-}$), see Methods]. $Tardbp$, $Ucp2$, $Nme6$, $Ndufaf1$, $Runx1$ showed differential expression levels similar to that of our sequencing data (Supplementary Fig. 6). We also tested different housekeeping genes for RT-qPCR normalization. Of the three housekeeping genes tested ($Gapdh$, $Actb$ and $H2az1$), $Gapdh$ produced the most consistent Ct values across 24 embryos tested (the standard deviation on Ct value of $Gapdh$, $Actb$ and $H2az1$ is 1.11, 5.33 and 1.52, respectively). The fact that more transcripts are downregulated than upregulated upon $Ddx1$ knockout suggests that DDX1's role is related to mRNA stability or biogenesis.

We were particularly intrigued by $Tardbp$ being downregulated by 5X $Ddx1$-low embryos as it encodes TDP-43 which is associated with alternative splicing and RNA processing[35]. It

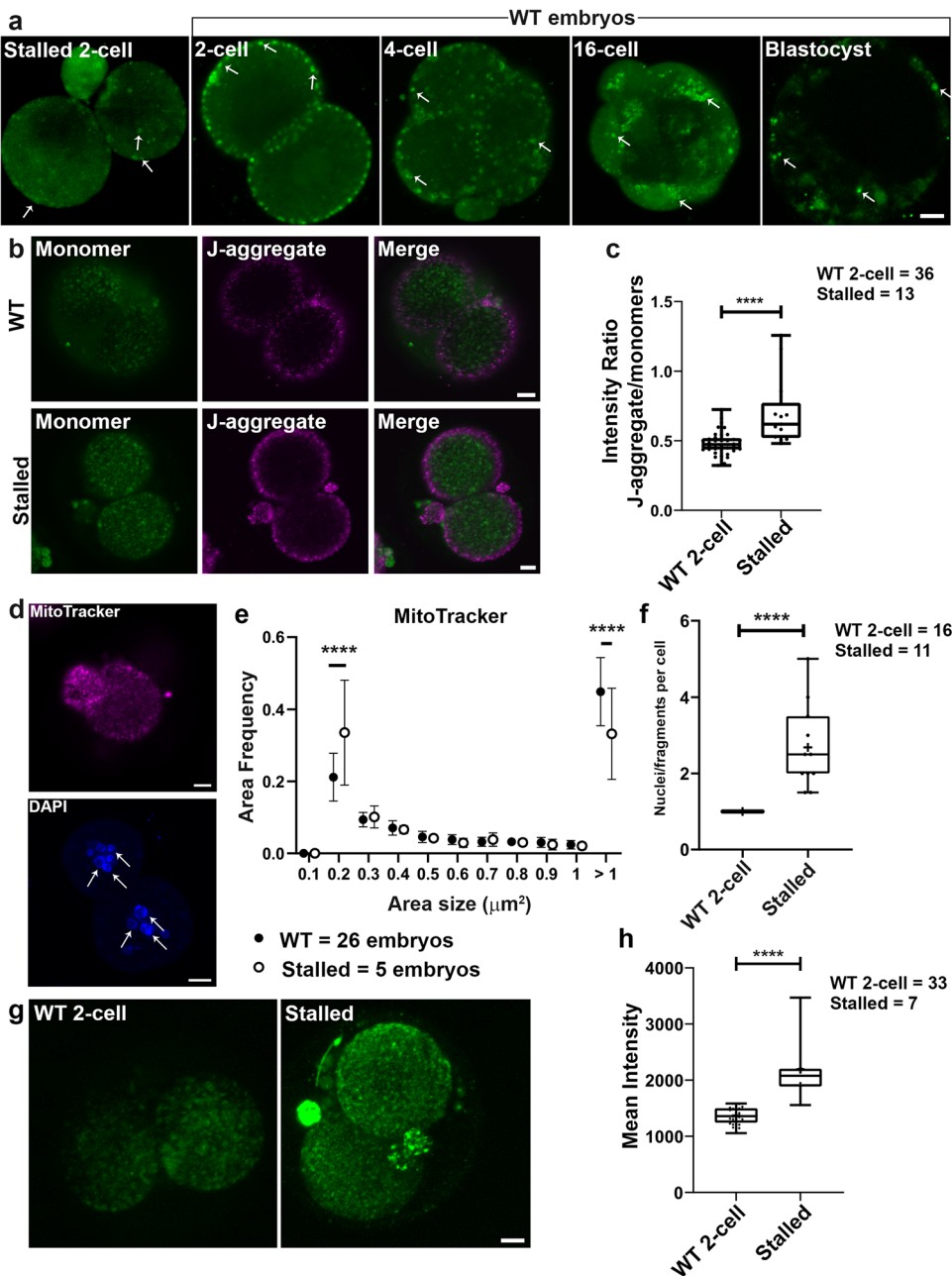

will be interesting to see whether TDP-43, along with cytoplasmic polyadenylation factors CPSF2 and CPEB1[36] (Supplementary Fig. 7a–c), co-compartmentalize with DDX1 in early stage embryos. A possible role for MARVs in cytoplasmic splicing and/or polyadenylation will be the subject of future studies.

It is noteworthy that the appearance of MARVs changes dramatically as the embryos mature, becoming larger and denser, finally encircling the nuclei of blastocysts. Fluo-4 AM staining shows that $Ca^{2+}$ forms larger microdomains as the embryos progress from 2-cell towards 16-cell compaction stages (Fig. 5a). We were not able to get valid data from direct Fluo-4 AM staining of the trophectoderm and inner cell mass in post-compaction embryos likely due to the surface trophectoderm absorbing most of the Fluo-4 AM dye. As an alternative approach, we first stained the embryos at the morula stage and cultured the embryos for an additional 24 h. The $Ca^{2+}$ staining pattern using this approach generated similar results to those shown in Fig. 5a. As DDX1 is an RNA helicase

that alters RNA structure in an ATP/ADP-dependent manner[37,38], and MARVs regulate $Ca^{2+}$ which is required to stimulate the mitochondria TCA cycle and oxidative phosphorylation for ATP production[39,40], these development-dependent changes in MARVs may reflect the need for increased local concentration of $Ca^{2+}$ to stimulate the production of ATP required for modulating DDX1 RNA binding/remodeling function.

## Discussion

Early mouse embryonic development relies heavily on maternal RNAs and proteins, with only a few zygotic genes identified to date that cause embryonic lethality at the 1–4 cell stages. These early embryonic lethality genes include *Ddx1*, suggesting a key role for DDX1 in maternal RNA/protein-related events associated with early embryonic development. Here, we show that DDX1 in early-stage embryos is located in membrane-bound vesicles that cluster in ring-like structures that we have named MARVs.

**Fig. 5 Loss of Ddx1 leads to increased mitochondrial membrane potential and mitochondrial ROS.** One-cell embryos from *Ddx1* heterozygote crosses were cultured in M16 medium for 72 h. One-cell embryos from *Ddx1* wild-type crosses were cultured for up to 96 h. Images were captured with a Zeiss LSM710 confocal microscope. **a** Fluo-4 AM staining (green) of stalled 2-cell embryos and wild-type 2-cell, 4-cell, 16-cell and blastocyst embryos. Three independent wild-type crosses were used for each stage with the exception of blastocysts where 2 crosses were used. Similar results were obtained for all embryos at the same stage. Arrows point to Fluo-4 AM staining in embryos. **b** JC-1 staining of stalled and wild-type 2-cell embryos. Stalled embryos have a stronger J-aggregate (magenta)/monomer (green) signal ratio compared to wild-type embryos indicating that the mitochondrial membrane potential is higher in stalled embryos. Single plane images are shown to better illustrate high potential mitochondria distribution. **c** Statistical analysis of mitochondrial membrane potential in wild-type ($n = 36$) and stalled ($n = 13$) embryos. Mean intensity values were plotted with Prism. Statistical analysis was performed with two-sided Students' *t*-test. ****indicates $p < 0.0001$ (with the exact $p = 2.24E\text{-}05$). Center line, median; box limits, 25th and 75th percentiles; whiskers, minimum to maximum. Error bars represent standard deviation. **d** Mitochondrial (top) and nuclear (bottom) fragmentation detected in stalled embryos. Arrows point to fragmented nuclei. Maximum intensity projections of the Z-stack images are shown for DAPI (MitoTracker, magenta; DAPI, blue). **e** The reduced size of MitoTracker Deep Red aggregates suggests fragmentation of mitochondria. Data were plotted with Prism. Statistical analysis was performed with two-sided multiple *t*-test using the Holm-Sidak method, with alpha = 0.05. ****indicates $p < 0.0001$ (both *p* values were less than 1E-06). Each circle represents the mean value, and the error bars represent standard deviation. Solid circles represent wild-type controls; empty circles represent stalled embryos. **f** The number of nuclei (or nuclei fragments) per cell was quantified and plotted with Prism. Statistical analysis was performed with two-sided Students' *t*-test. ****indicates $p < 0.0001$ (with the exact $p = 1.92E\text{-}06$). Center line, median; box limits, 25th and 75th percentiles; whiskers, minimum to maximum. Error bars represent standard deviation. **g** Mitochondrial ROS in wild-type and stalled 2-cell embryos was detected using the MitoSOX dye (green). Maximum intensity projections of the Z-stack images at each stage are shown. **h** Data obtained from (**g**) were plotted with Prism. Statistical analysis was performed with two-sided Students' *t*-test. ****indicates $p < 0.0001$ (with the exact $p = 1.59E\text{-}08$). Center line, median; box limits, 25th and 75th percentiles; whiskers, minimum to maximum. Error bars represent standard deviation. Scale bars = 10 μm. Z-stack images of each embryo was used for the statistical analysis in (**c**), (**f**) and (**h**). Single plane images of each embryo (middle sections) were used for the statistical analysis in (**e**). WT = wild-type.

Electron microscopy of 2-cell embryos reveals a core of RNA within each membrane-bound vesicle. In addition to DDX1 and RNA, $Ca^{2+}$ and cytoplasmic polyadenylation factors have been identified in MARVs. Cellular compartmentalization of RNA and RNA processing/modifying proteins within membrane-bound structures suggests a need for their localized concentration or protection. The fact that these membrane-bound vesicles cluster together further suggests the need for enhancement of processes associated with the membrane-bound vesicles.

Treatment of 2-cell embryos with the inhibitor of mitochondrial oxidative phosphorylation, FCCP, has previously been shown to increase cytoplasmic $Ca^{2+}$ and disrupt $Ca^{2+}$ microdomains at the subplasmalemmal cytoplasm of 2-cell embryos[16]. Our results indicate that FCCP may cause the release of $Ca^{2+}$ from MARVs. The latter is analogous to what has been observed in somatic cells, except that in somatic cells, the endoplasmic reticulum (ER) is the $Ca^{2+}$ storage site instead of MARVs[41]. It is well known that ER-mitochondria interact with each other in somatic cells through the mitochondria-associated membrane (MAM) with multiple tethering complexes to ensure proper signal transduction between the two organelles[42]. Although both organelles play important roles in cellular $Ca^{2+}$ homeostasis, mitochondria often rely on ER $Ca^{2+}$ transfer through the MAM to stimulate oxidative phosphorylation and the TCA cycle for ATP production or to control cell death through autophagy or apoptosis[43]. $Ca^{2+}$ transfer between ER and mitochondria is mainly through the inositol triphosphate receptor located on the ER membrane and its interaction with VDAC, the voltage-dependent anion channel located in the mitochondrial outer membrane. After passing through the outer membrane, the $Ca^{2+}$ is taken up by the mitochondrial calcium uniporter (MCU) located in the inner mitochondrial membrane[42]. Notably, the outer mitochondrial membrane is permeable to small molecules <10 kDa, in contrast to the inner membrane which is completely impermeable even to small molecules (except for $O_2$, $H_2O$ and $CO_2$)[44]. Therefore, $Ca^{2+}$ can pass through the outer membrane while following an ion gradient without VDAC, but can only pass through the inner mitochondrial membrane via the MCU. Thus, proximity, along with an ion gradient, may be important factors in controlling $Ca^{2+}$ transfer from ER to mitochondria. Unlike the close connection between ER and mitochondria which are typically located within 100 nm of each other in somatic cells[44], MARVs and mitochondria are much further apart (>200 nm) indicating that MARVs are unlikely to be close enough to mitochondria to form a complex for $Ca^{2+}$ transfer through the VDAC tunnel. In agreement with this, a recent study has demonstrated that although MCU is present in oocyte mitochondria, VDAC appears to be absent from oocyte mitochondria[45]. This property of oocyte mitochondria may be carried over to pre-blastocyst embryos. We propose that the local high concentration of $Ca^{2+}$ in MARVs allows $Ca^{2+}$ transfer from MARVs to mitochondria through the mitochondria outer membrane following an ion gradient.

Mitochondria from oocyte to pre-blastocyst stage have a distinct morphology compared to mitochondria found in somatic cells. These mitochondria have been described as "immature" as they are smaller in size, more electron dense, have fewer numbers of cristae, and consume less oxygen compared to regular mitochondria[11,46]. Oocyte mitochondria are abundant but function suboptimally to control the production of toxic mtROS[45]. In spite of their "immature" mitochondria, early embryos rely heavily on mitochondria for their metabolic needs[11,47]. With $Ca^{2+}$ being a main stimulator of the TCA cycle and oxidative phosphorylation for energy production, $Ca^{2+}$ distribution needs to be tightly controlled in order to regulate mitochondria function[39,40]. Thus, the normal $Ca^{2+}$ release from ER to mitochondria may not be suitable for early-stage embryos. Although more work needs to be done, we propose that MARVs provide localized and more precise release of $Ca^{2+}$ for optimal mitochondria function during early embryonic development.

Consistent with our proposed role for MARVs in $Ca^{2+}$ segregation, disruption of MARVs in $Ddx1^{-/-}$ embryos results in stalled 2- to 4-cell stage embryos with disrupted $Ca^{2+}$ distribution, upregulation of mitochondrial membrane potential, increased ATP levels and mtROS production, and nuclear and mitochondrial fragmentation. Although we do not know the sequence of these events, we postulate that disruption of $Ca^{2+}$ is the root cause for this series of events. As mentioned above, $Ca^{2+}$ is needed to stimulate mitochondrial ATP production which is accompanied by mtROS production. Therefore, a change in $Ca^{2+}$ balance could have a profound effect on embryonic development. For example, although mtROS is essential for embryonic development[48],

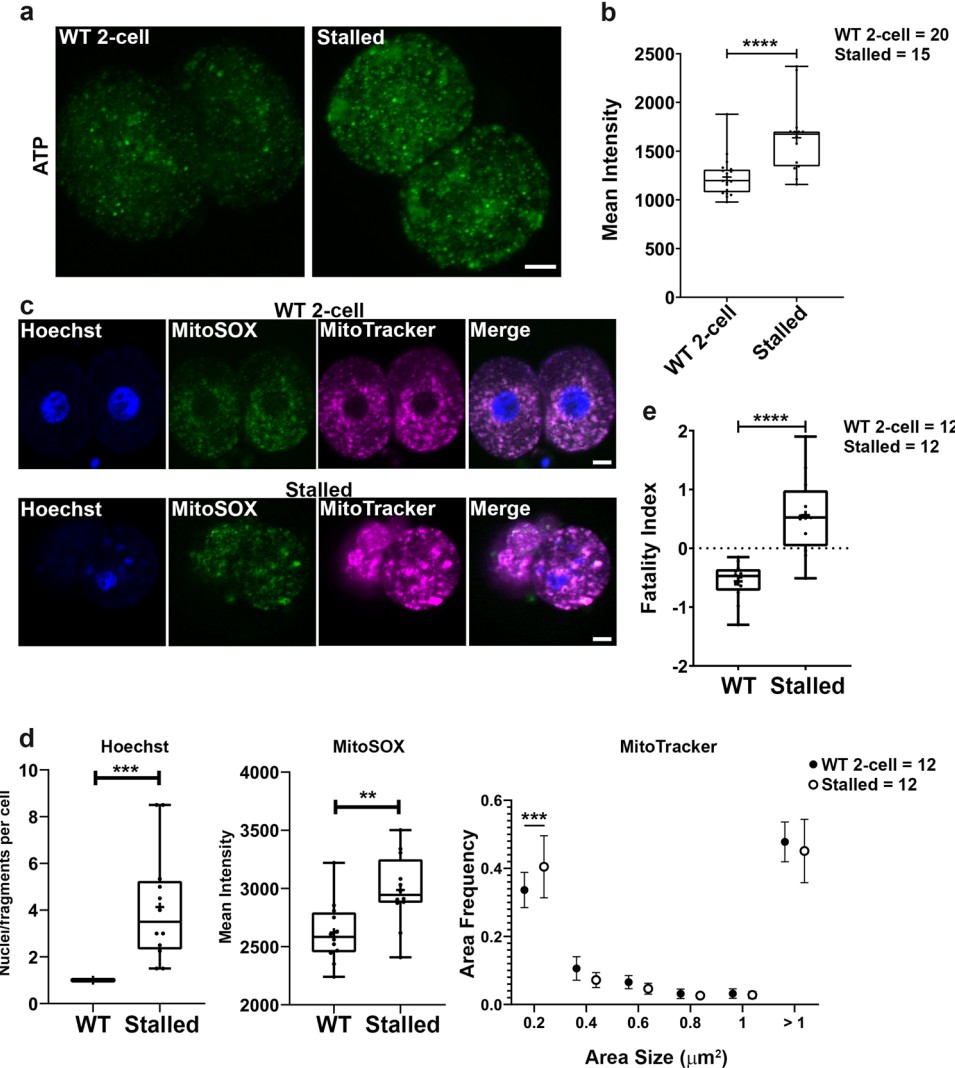

**Fig. 6 Loss of Ddx1 leads to higher ATP levels and a higher fatality index.** One-cell embryos from *Ddx1* heterozygote crosses were cultured for 72 h. One-cell embryos from *Ddx1* wild-type crosses cultured to the 2-cell stage were used for comparison. **a** BioTracker ATP-red staining (green) of stalled 2-cell embryos show higher ATP levels compared to wild-type 2-cell embryos. Maximum intensity projections of the Z-stack images at each stage are shown. **b** Statistical analysis of (**a**) using Z-stack images for each embryo. Mean intensity values were plotted with Prism. Statistical analysis was performed with two-sided Students' *t*-test. ***indicates *p* < 0.001 (with the exact *p* = 1.34E-04). Center line, median; box limits, 25th and 75th percentiles; whiskers, minimum to maximum. Error bars represent standard deviation. **c** Triple staining of nuclei (Hoechst 33342) (blue), mtROS (MitoSOX) (green) and mitochondria (MitoTracker) (magenta) in stalled versus wild-type two-cell embryos. **d** Stalled embryos have higher mtROS and more fragmented nuclei and mitochondria based on statistical analysis of each channel in (**c**) plotted individually. Statistical analysis was performed with either two-sided Students' *t*-test or two-sided multiple *t*-test using the Holm-Sidak method, with alpha = 0.05. ***indicates *p* < 0.001. **indicates *p* < 0.01. Exact *p* value for left panel is 0.0002, center panel is 0.0045, and right panel is 0.00045. For the box plots, center line, median; box limits, 25th and 75th percentiles; whiskers, minimum to maximum. Error bars represent standard deviation. For the symbol plot, each circle represents the mean value, and the error bars represent standard deviation. Solid circles represent wild-type controls; empty circles represent stalled embryos. **e** Statistical analysis of fatality index constructed with weighted average index construction using the *z*-scores of nuclei (or nuclei fragments) number/cell, MitoSOX intensity and percentage of mitochondria with size <0.2 μm², and plotted with Prism. Statistical analysis was performed with two-sided Students' *t*-test. ****indicates *p* < 0.0001 (with the exact *p* = 2.66E-05). Center line, median; box limits, 25th and 75th percentiles; whiskers, minimum to maximum. Error bars represent standard deviation. Scale bars = 10 μm. Numbers of embryos analysed are indicated on the right hand side. WT = wild-type. Z-stack images were used for statistical analysis of nuclei and mtROS whereas single plane images of each embryo (middle sections) were used for statistical analysis of mitochondria.

overproduction of mtROS can lead to oxidative stress and embryonic lethality[47,49]. Thus, aggregates of individual vesicles in MARVs may be more advantageous for early embryonic functions than the interconnected ER, as they allow a more precise control of $Ca^{2+}$ distribution in embryos. In summary, we propose that MARVs control the spatial distribution of $Ca^{2+}$ within early-stage embryos such that mitochondria activity is precisely regulated to maximize ATP production while controlling ROS production and minimizing ROS-induced damage to the embryos.

In addition to $Ca^{2+}$, electron microscopy reveals an RNA core in the membrane-bound vesicles that form the ring-like MARV structures. As RNA compartmentalization is often achieved by liquid-liquid phase separation, the need for a membrane-bound organelle to enclose the RNA cores is not clear. In this regard, it is important to note that the RNA cores themselves may form through liquid-liquid phase separation. Further segregation of these RNAs within membrane-bound vesicles may be important for protection of the RNAs and/or quick access to the RNAs when

required. Proximity of the RNA core to proteins found in the membrane-bound vesicles may therefore optimize the utilization of these RNAs.

Oocytes store large quantities of maternal RNAs that are critical for early mouse development[50]. RNA polyadenylation and deadenylation are processes by which translation is controlled in oocytes[36]. mRNAs with short poly(A)-tail are not translated or only poorly translated, with polyadenylation increasing translation efficiency. Oocyte growth is accompanied by the appearance of subcortical aggregates in the cytoplasm. The presence of cytoplasmic polyadenylation factor CPEB1 in these subcortical aggregates suggest a role in cytoplasmic polyadenylation during oocyte growth and maturation[51]. However, subcortical aggregates have disappeared by the MII phase of oocyte maturation, so are no longer present during the third and last wave of cytoplasmic polyadenylation that takes place after fertilization[52]. We thus propose that the control of maternal RNA translation associated with the third wave of polyadenylation either occurs in MARVs or involves MARVs. Proximity of RNAs to the RNA binding/unwinding DDX1 may allow remodelling of RNAs, making them accessible to cytoplasmic polyadenylation factors CPSF2 and CPEB1, all of which are found in MARVs. $Ca^{2+}$ may also be involved in polyadenylation as it's been associated with activation and phosphorylation of CPEB1[12]. RNA sequencing of single embryos from heterozygote crosses has revealed 300 down-regulated and 90 up-regulated genes in embryos with low levels of *Ddx1* RNA compared to embryos with elevated levels of *Ddx1* RNA. Some of the down-regulated genes are associated with developmental processes and mitochondrial function. Future work will involve determining whether the differentially expressed transcripts identified by RNA sequencing are indeed found in MARVs and whether there is interplay between these transcripts, $Ca^{2+}$ signaling and mitochondrial function.

In summary, we have discovered a membrane-bound organelle within the cytoplasm of early-stage embryos that contains DDX1 and RNAs, as well as $Ca^{2+}$. These organelles aggregate to form ring-like structures that we have called MARVs. We propose that MARVs have evolved to either allow retention of RNAs and $Ca^{2+}$ to highly localized regions of the developing embryo or provide a barrier to entry of other molecules. Our results using stalled embryos from *Ddx1* heterozygote crosses indicate that MARVs may be important for the timed release of $Ca^{2+}$ required for mitochondrial energy production (Fig. 7). On the other hand, our single embryo RNA sequencing analysis indicates that DDX1 affects a subset of genes required for zygotic genome activation, cytoplasmic polyadenylation as well as RNA degradation (such as *Mettl14* and *Tent5a*). As the major wave of zygotic genome activation is essential in 2-cell mouse embryos for proper embryonic development, it is possible that MARVs may also play essential roles in the maternal to zygotic transition process. On a practical level, and in consideration of their role in activating mitochondria, a better understanding of MARVs may lead to additional approaches to rescue aged oocytes with under-performing mitochondria which contributes to reduced fertility[13].

## Methods

**Embryo and oocyte collection**. All embryos used in our experiments were obtained under natural-mating conditions. For embryo cultures, M16 medium (Sigma–Aldrich) was allowed to equilibrate overnight in a 37 °C incubator with 5% $CO_2$. The generation of the *Ddx1* mouse knockout line in the FVB/N background has been described previously[53]. For timed pregnancies, females were tested for the presence of vaginal plugs, with plugged date labelled as embryonic day (E) 0.5. For collection of E0.5 to E2.5 embryos, oviducts were removed and placed in 35 mm tissue culture dishes where they were flushed with equilibrated (37 °C) M16 medium using blunt-ended needles. E0.5 embryos were treated with 300 µg/ml hyaluronidase (Sigma–Aldrich H4272; 750–3000 unit/mg) for the removal of cumulus cells. For collection of E3.5 embryos, the uterus was removed and placed in 35 mm tissue culture dishes. Uterine horns were flushed with equilibrated

(37 °C) M16 medium using 18 G needles. All embryo transfers and manipulations after flushing were done with capillary tubes (World Precision Instruments).

To obtain meiosis II arrested (MII-arrested) oocytes, FVB/N wild-type mice were injected with 5 IU pregnant mare serum gonadotropin (MSD Animal Health) followed by 5 IU human chorionic gonadotropin (MSD Animal Health) 48 h later. The oviducts were then removed and flushed with M16 medium using blunt-ended needles. MII stage oocytes were treated with 300 µg/ml hyaluronidase for the removal of cumulus cells. All animal experiments were carried out in accordance with the approved guidelines of the Cross Cancer Institute Animal Care Committee (protocol #AC20253).

**Culturing embryos**. M16 medium was allowed to equilibrate overnight in a 37 °C incubator with 5% $CO_2$. The humidified culture chamber was an adaption from Gasperin et al.[54]. Briefly, all gaps between each well of 96-well plates were filled with 150 µl autoclaved ddH$_2$O and the centermost 3*3 wells were filled with 50 µl/well of equilibrated M16 medium. After the embryos were collected, they were washed three times in the first 3 wells of M16 medium and placed in the center well of the 3*3 matrix and cultured for the designated times in a 37 °C incubator with 5% $CO_2$.

**Primary antibody conjugation**. The specificity of the anti-DDX1 antibody used for our experiments has been previously verified[55], and used for mouse embryo immunostaining[8,53]. For direct conjugation of anti-DDX1 antibody to Atto 550, one ml of anti-DDX1 antibody was IgG-purified with a Melon Gel IgG Spin Purification Kit (Thermo Fisher) and concentrated to 1 mg/ml. The purified anti-DDX1 antibody was then conjugated to Atto 550 using the Atto 550 Protein Labeling Kit (Sigma-Aldrich, Cat #51146) following the manufacturer's protocol.

**Immunofluorescence staining**. Embryos were flushed from oviducts using PBS or transferred from culture M16 medium to PBS and fixed in 4% paraformaldehyde for 15 min. The embryos were then washed three times with PBS + 0.01% Tween (PBST) and permeabilized with PBS + 0.5% Triton-X-100 for 15 min. Next, the embryos were washed three times in PBST and incubated in 0.5% fish gelatin blocking solution for 30 min, and then incubated for 1 h in PBST-diluted primary antibodies. The embryos were washed 3 times in PBST, followed by 1 h incubation with PBST-diluted secondary antibodies (if secondary antibodies were used). After washing three times in PBST, embryos were placed on a slide and mounted with Mowiol (Calbiochem) containing 4′,6-diamidino-2-phenylindole (DAPI). For CPEB1, the embryos were immunostained following a previously published protocol with slight modifications[56], including fixation in 4% paraformaldehyde for 15 min, washes in 0.1% PBST and blocking in 10% BSA in PBS. The following primary antibodies were used: rabbit anti-DDX1 (1:800 batch 2910 made in house), mouse anti-RPS6 (1:50 Santa Cruz, Cat #sc-13007), mouse anti-MRPS27 (1:100 Santa Cruz, Cat #sc-390396), mouse anti-MRPL42 (1:100 Santa Cruz, Cat #sc-515820), mouse anti-MRPL44 (1:100 Santa Cruz, Cat #sc-515503), mouse anti-CPEB1 (1:50 Santa Cruz, Cat #sc-514688), and goat anti-CPSF2 (1:100 Santa Cruz, Cat #sc-26658). The secondary antibodies used were donkey anti-mouse Alexa 647 (1:400 Molecular Probes, Thermo Fisher), donkey anti-rabbit 555 (1:400 Molecular Probes, Thermo Fisher), and donkey anti-goat 647 (1:400 Molecular Probes, Thermo Fisher).

For super resolution microscopy, embryos were imaged using a Leica TCS SP8 Falcon STED super resolution microscope with a 100X/NA1.4 oil lens to capture single sections. As whole mount embryos are relatively thick, the embryos were imaged at the sections that were the closest to the cover slip. For confocal Z-stacks, the embryos were imaged with a Leica TCS SP8 microscope and 100X/NA1.4 oil lens. The Leica TCS SP8 microscope was operated with Leica Application Suite X, version 3.5.

A subset of immunostained embryos was imaged using a Zeiss Laser Scanning Confocal microscope (LSM710) mounted on an inverted microscope (Axio Observer). The images taken with LSM710 were operated with Zeiss software suite (ZEN, Version 2011). Embryos were mounted with Mowiol (Calbiochem) containing DAPI. Images were taken with a 40X/NA1.3 oil lens[8].

**Inhibition of global and mitochondrial translation**. M16 medium (50 µl) containing either 150 µg/ml cycloheximide (Sigma–Aldrich) or 200 µg/ml chloramphenicol (Sigma–Aldrich) was used for global or mitochondrial translation inhibition, respectively. Oviducts were flushed with equilibrated M16 medium on E0.5 (at 10:00) and embryos were treated with 300 µg/ml hyaluronidase to remove the cumulus cells. The embryos were then washed three times in equilibrated M16 medium and incubated at 37 °C until 16:00. The embryos were then transferred to M16 medium with cycloheximide or chloramphenicol and incubated at 37 °C for 8 h, followed by fixation in 4% paraformaldehyde and immunostaining. Control embryos were incubated in M16 medium without inhibitors.

**Saponin treatment**. E1.5 embryos were collected by oviduct flushing and washed three times in equilibrated M16 medium. Embryos were incubated in equilibrated M16 medium containing 0.1% saponin for 6 min at room temperature. Control embryos were incubated in equilibrated M16 medium without saponin (Sigma–Aldrich). The embryos were washed three times in equilibrated M16

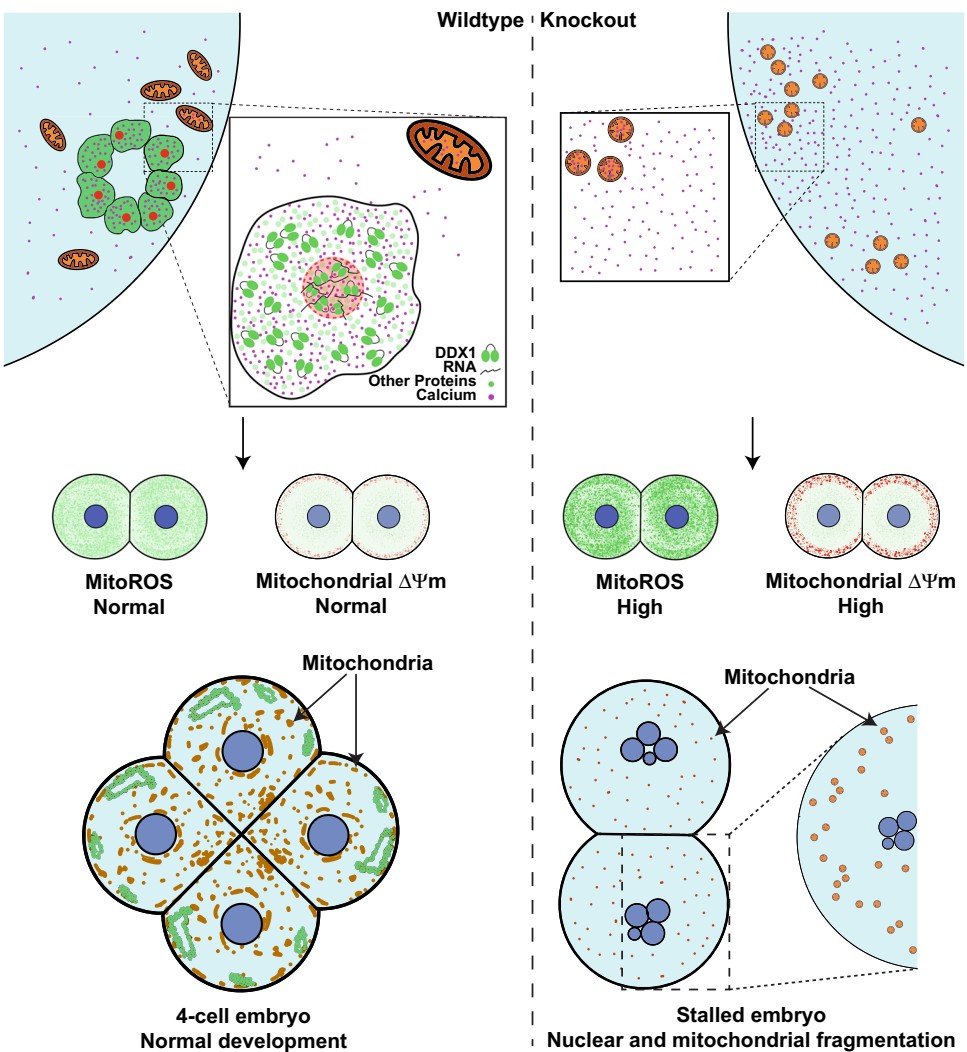

**Fig. 7 Schematic depiction of DDX1 and MARV function.** Wild-type embryos develop normally when MARVs are present. However, absence of MARVs in *Ddx1* knockout embryos results in disrupted $Ca^{2+}$ distribution which causes upregulation of mitochondrial activity and increased mitochondrial ROS, leading to nuclear and mitochondrial fragmentation.

medium and incubated in equilibrated M16 medium for an additional 30 min and processed for live cell imaging, immunofluorescence staining or immunogold labelling.

**Live cell imaging.** E0.5 or E1.5 embryos were washed three times in equilibrated M16 medium. E0.5 embryos were treated with 300 μg/ml hyaluronidase to remove the cumulus cells. For JC-1, MitoSOX, ER-Tracker, MitoTracker Deep Red and BioTracker ATP-red live cell staining, embryos were incubated for 30 min in equilibrated M16 medium containing 10 μg/ml JC-1 dye (Thermo Fisher), 2 μM MitoSOX (Thermo Fisher), 200 nM ER-Tracker Red (Thermo Fisher), 100 nM MitoTracker Deep Red (Cell Signaling Technology), or 15 min in 10 μM Bio-Tracker ATP-red (Millipore). The embryos were washed three times in equilibrated M16 medium and transferred to ultrathin bottom 96-well plates (Grenier bio-one Cat. 6550900) containing 50 μl equilibrated M16 medium or chambered coverglass (Sigma–Aldrich) containing 400 μl equilibrated M16 medium, and processed for imaging using a LSM710 confocal microscope. For triple color staining of Hoechst 33342, MitoSOX and MitoTracker Deep Red, embryos were incubated in equilibrated M16 medium containing 1 μg/ml Hoechst 33342 (Thermo Fisher), 2 μM MitoSOX (Thermo Fisher) and 100 nM MitoTracker Deep Red (Cell Signaling Technology) for 30 mins. The embryos were washed three times in equilibrated M16 medium and transferred to chambered coverglass (Sigma–Aldrich) containing 400 μl equilibrated M16 medium and processed for imaging using a LSM710 confocal microscope. For $Ca^{2+}$ staining, the embryos were incubated for 30 min in equilibrated M16 medium containing 5 μM Fluo-4 AM dye. The embryos were washed three times in equilibrated M16 medium, followed by incubation in 50 μl of equilibrated M16 medium for 30 min to remove non-specifically-bound dye. For live cell co-staining with other dyes, the embryos were transferred to equilibrated M16 medium containing the dye of interest for 30 min and imaged in ultrathin

bottom 96-well plates containing 50 μl equilibrated M16 medium. For saponin treatment, the embryos were incubated in equilibrated M16 medium with or without 0.1% saponin before incubation with MitoTracker Deep Red. Live cell imaging was carried out using a Zeiss LSM710 laser scanning microscope and a 20X/NA0.8 lens. All LSM710 images were taken with Zeiss software suite (ZEN 2011).

**Treatment with the protonophore FCCP and $Ca^{2+}$ chelator EGTA-AM.** 2-cell mouse embryos were collected by oviduct flushing at E1.5, followed by three washes in equilibrated M16 medium. Embryos were transferred to equilibrated M16 medium containing 5 μM FCCP (Sigma–Aldrich) for 2 h or 100 μM EGTA-AM (Millipore) for 3 h. An equal amount of DMSO was used as control. Embryos were then washed three times in equilibrated M16 medium and processed for immunofluorescence staining.

**Calcium free embryo cultures.** $Ca^{2+}$ containing regular M16 medium and $Ca^{2+}$ free M16 medium were prepared according to the CSH protocol[57]. E1.5 embryos obtained from oviduct flushing were collected and washed three times in either equilibrated regular M16 or $Ca^{2+}$ free M16 medium. Embryos were then transferred to their designated medium ($Ca^{2+-}$ containing or $Ca^{2+}$ free M16 medium) and cultured for 24 h. Embryos were then processed for $Ca^{2+}$ live cell imaging using Fluo-4 AM or DDX1 immunofluorescence staining.

**Co-staining of calcium and DDX1.** Fluo-4 AM stained 2-cell mouse embryos were fixed and permeabilized using −20 °C methanol-acetone (1:1) for 5 min at 4 °C, followed by Triton-X 100. Methanol-acetone fixed embryos were then incubated in PBST (10 min X3) for rehydration followed by incubation with Atto 550-

conjugated anti-DDX1 antibody. Stained embryos were placed in ultrathin bottom 96-well plates (Grenier bio-one Cat. 6550900) and imaged immediately using a Zeiss LSM710 microscope.

**Single embryo RNA expression analysis**. Eighteen embryos collected by oviduct flushing of two E0.5 heterozygous *Ddx1* crosses obtained by natural mating. Embryos were then treated with Acidic Tyrode's solution (Sigma–Aldrich) for zona removal and cultured in equilibrated M16 for 24 h. Individual embryos ($n = 18$) were collected and frozen in liquid nitrogen until further use. A sequencing library for each embryo was prepared using the NEBNext Single Cell/Low Input RNA Library Prep Kit (E6420S) following the manufacturer's protocol. The sequencing libraries were then submitted to MedGenome Labs Ltd. (Foster City, CA) for next generation sequencing. The raw sequencing data were then aligned with STAR aligner and counted by featurecounts R. *Ddx1* high versus *Ddx1* low embryos were determined based on the number of *Ddx1* transcripts per million transcripts (TPM) where *Ddx1* TPM levels ≤0.6 or >3 are defined as low or high levels of *Ddx1*, respectively. The mRNA profile of two separate groups of embryos (*Ddx1* high vs *Ddx1* low) were then analyzed by DESeq2 package in R. *P*-value of each gene was obtained by Wald test and corrected with Benjamini and Hochberg method. Genes with FDR < 0.1 were selected.

In order to validate our sequencing results, we extracted RNAs from an independent batch of 24 two-cell embryos generated from heterozygous *Ddx1* crosses. The RNAs were reverse transcribed and amplified with the NEBNext Single Cell/Low Input cDNA Synthesis & Amplification Module (E6421S) following the manufacturer's protocol. Three high *Ddx1* and three low *Ddx1* embryos were selected for validation based on their *Ddx1* mRNA levels. The selected transcripts were quantified using RT-qPCR in these six embryos (high *Ddx1* vs low *Ddx1*). Primers for selected transcripts (Supplementary Dataset 2) were used for RT-qPCR analysis using 1 µl of diluted cDNA per 10 µl reaction containing BrightGreen 2X-qPCR MasterMix-ROX (Applied Biological Materials). PCR was performed in 96-well plates using the QuantStudio 6 Flex Real-Time PCR system (Applied Biosystems). The run method was customized as follows: Pre-PCR (Ramp rate 1.9 °C/s, 95 °C for 10 min), 40 cycles PCR (Ramp rate 1.9 °C/s, 95 °C 15 s, Ramp rate 1.9 °C/s, 58 °C 30 s, Ramp rate 1.6 °C/s, 72 °C 1 min). The mRNA levels were normalized to *Gapdh* levels with Excel and plotted with Prism (GraphPad). Log$_{10}$ scaled levels are shown (Supplementary Fig. 6). The multiple *t*-test with Holm-Sidak method was used to test for significance between low *Ddx1* vs high *Ddx1* embryos.

**Imaging analysis**. Images were imported and analysed in ImageJ. Mean intensity values of different channels were used for JC-1, MitoSOX and ATP statistical analysis. Coloc2 ImageJ colocalization analysis plugin was used for the analysis of mitochondria and Ca$^{2+}$ with and without saponin treatment. Pearson correlations for each image were marked for statistical analysis with Prism (GraphPad). The unpaired *t*-test was carried out using Prism (GraphPad). JC-1, MitoSOX and ATP data were plotted with box-and-whisker plots. Their statistical significances were evaluated with Student's *t*-test. For all box-and-whisker plots, elements are defined as below: center line, median; box limits, 25th and 75th quartiles; whiskers, min to max with all data points shown.

For nuclear fragmentation analysis, each nucleus was manually labelled. Data were then plotted with box-and-whisker plot in Prism (GraphPad). Students' *t*-test was used for statistical significance calculation.

For mitochondria fragmentation analysis, Particle Analysis plug-in in ImageJ was used. The images were first thresholded with Auto Local Threshold. Particles with size of 10 pixel$^2$ and above were recorded. Counted particles within different size ranges were summarized and converted to µm$^2$. Data were then plotted with Prism (GraphPad) as interleaved symbols, with mean values shown as circles and error bars representing SD. The multiple t-test with Holm-Sidak method was used to test for significance.

For analysis of size of MARVs, 3D object counter on GPU plug-in[58] in ImageJ was used. The background of z-stack images was subtracted, and the images were thresholded with Auto Thresholding. 3D objects with sizes ≥10 voxels were recorded. Counted objects within different size ranges were summarized and converted to µm$^3$. Data were then plotted with Prism (GraphPad) as described for mitochondrial fragmentation.

For co-staining with Hoechst 33342, MitoSOX and MitoTracker Deep Red, each channel was calculated individually as stated above. Only particles less than 0.2 µm$^2$ were used for MitoTracker channel. The *z*-scores of these 3 channels were averaged to construct a new index using weighted average index construction. The significance between the wild-type and the stalled embryos was calculated with Students' *t*-test.

**Immunogold labelling**. Ultralow melt agarose (SeaPrep) (4%) was prepared and melted at 60 °C, then placed in a 37 °C water bath. Embryos at different developmental stages were collected and fixed in 4% paraformaldehyde in 0.1 M phosphate buffer for 1 h at room temperature. The embryos were transferred to a 250-µl tube and immersed in 5 µl of 4% agarose. The embryos were placed at 4 °C to allow the agarose to solidify.

The embryos were rinsed in 0.1 M phosphate buffer pH 7.3. The embryos were dehydrated in series of pre-cooled (4 °C) ethanol solutions (30%, 50% and 70% 15 min each and 95% and 100% ETOH 10 min each). For embedding, the embryos were placed in LR white resin-100% ethanol (2:1) for 2 h at 4 °C and pure LR white resin overnight. LR white was polymerized under UV for 96 h at 4 °C. Samples were cut into 100 nm sections with a Leica EM UC6 ultramicrotome and the sections were collected on 400 mesh nickel grids.

For immunogold labeling, the grids were immersed in 50 mM NH$_4$Cl for 1 min at room temperature and then blocked with 0.5% fish gelatin in PBS for 45 min at room temperature. The grids were incubated with anti-DDX1 antibody (1:1,000 dilution) in PBS for 3 h at room temperature, followed by 10 washes in PBS. The grids were then incubated in 1:100 diluted goat anti-rabbit 10 nm nano-gold (Electron Microscopy Sciences, Cat #25109) in PBS for 1 h. The grids were washed in 0.05% fish gelatin in PBS, followed by PBS. The samples were fixed again in 2% paraformaldehyde +2.5% glutaraldehyde for 15 min at room temperature, rinsed in H$_2$O and treated with 1% OsO$_4$ for 10 min at room temperature. The samples were washed in H$_2$O and stained with 1% uranyl acetate for 25–30 min at room temperature and 1% lead citrate for 3 min at room temperature. Images were obtained with a JEOL JEM-2100 transmission electron microscope at 200 kV. TEM images were acquired with Gatan Microscope Suite Software.

**OsO$_4$/uranyl acetate staining**. The embryos were fixed in 2% paraformaldehyde +2.5% glutaraldehyde in 0.1 M phosphate buffer (pH 7.3) for 1 h at room temperature and placed in low melt agarose as described under immunogold labelling, washed in 0.1 M phosphate buffer pH 7.3, and post-fixed in 0.1 M phosphate buffer containing 1% OsO$_4$ for 35 min at room temperature. The embryos were then incubated in freshly made 1% carbohydrazide solution (in H$_2$O) at room temperature for 5 min and quickly rinsed with ddH$_2$O. The embryos were incubated again in 0.1 M phosphate buffer with 1% OsO$_4$ for 15 min and washed with ddH$_2$O.

The samples were dehydrated in 30% and 50% ethanol solutions (10 min each). The samples were then further dehydrated in 70% ethanol overnight at room temperature (if uranyl acetate was used, 1% uranyl acetate was added to the 70% ethanol), followed by 95% ethanol for 10 min and 3 × 100% ethanol (10–15 min each). A series of resin infiltrations were performed with Spurr's resin [Spurr's:100% ethanol (1:4) for 1 h and 40 min, Spurr's:100% ethanol (1:1) for 1 h and 35 min, Spurr's:100% ethanol (3:1) for 2 h]. The samples were then embedded in pure Spurr's resin for 1 h, with two more changes into fresh pure Spurr's resin overnight and then 2 h. The infiltrated embryos were polymerized in a 65 °C oven for 21 h. The samples were then cut into 100 nm sections. If uranyl acetate was used in the experiment, samples were final-stained in 1% uranyl acetate for 15 min followed by 1% lead citrate for 4 min. TEM images were acquired with Gatan Microscope Suite Software (Digital Micrograph, Version 3.23).

**Energy filtered transmission electron microscopy (EFTEM)**. A Gatan energy filter system (Quantum) (Gatan, CA, USA) mounted on a JEOL TEM (JEM 2100) with a Lab6 filament was used for EFTEM elemental mapping of phosphate and nitrogen. For phosphate mapping, embryos were stained with OsO$_4$ (no uranyl acetate) to visualize the MARVs and analysed using three window EFTEM. Pre-edge images were obtained at 122 eV and 127 eV, and the post-edge image was obtained at 155 eV. The energy-selecting window was set at 5 eV. For nitrogen mapping, embryos were DDX1 immunogold-labelled to identify MARVs. No OsO$_4$/uranyl acetate was used in nitrogen mapping to avoid interference of osmium with nitrogen. Pre-edge images were obtained at 368 eV and 388 eV, and the post-edge image was obtained at 411 eV. The energy-selecting window was set at 20 eV. EFTEM images were acquired with Gatan Microscope Suite Software.

**Statistics and reproducibility**. Statistical analysis used for each experiment is described in each of the figure legends. Each datapoint represents an individual embryo. No statistical method was used to predetermine sample size and no data were excluded from the analyses. The mice treated with different treatments or controls were allocated randomly. The intensity analyses and particle size analyses were calculated blindly to obtain the values for each sample. These values were then used for statistical analysis.

**Reporting summary**. Further information on research design is available in the Nature Research Reporting Summary linked to this article.

## Data availability

Sequencing data have been uploaded to the NCBI Sequence Read Archive and can be accessed under the accession numbers "SRR17058908", "SRR17058907", "SRR17058906", "SRR17058905", "SRR17058904", "SRR17058903". All other data needed to reproduce and evaluate the conclusions in this paper are present in the main text and supplementary data or from the corresponding author upon reasonable request. Original data of all figures presented in the main text and supplementary data are provided in the source data file. *Ddx1* heterozygous mice are available upon request. Source data are provided with this paper.

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

## Acknowledgements

We thank Dan McGinn and Cheryl Santos for their help with the mouse experiments. We are also grateful for the support of Gerry Barron, Guobin Sun and the Cross Cancer Institute Cell Imaging Facility. This work was supported by a grant from the Canadian Institutes of Health Research – grant number 162157 to R.G.

## Author contributions

Y.W. and R.G. designed the study; Y.W. and L.Y. performed embryo collections and embryo cultures; Y.W. performed immunofluorescence staining; L.Y. carried out the breeding experiments for generation of embryos; L.L. performed RNA sequencing library preparation and RT-qPCR validation; Y.W. performed RNA sequencing data analysis; Y.W. and X.X. performed confocal and STED microscopy experiments; P.G., Y.W. and X.S. performed TEM sample preparation, electron microscopy observation and EFTEM; and Y.W. and R.G. wrote the paper. All authors reviewed the paper.

## Competing interests

The authors declare no competing interests.
