## [Peer Review File · Nature Communications]

REVIEWER COMMENTS

Reviewer #1 (Remarks to the Author):

The paper “DDX1 Vesicles Control Calcium-dependent Mitochondrial Activity in Embryos” by Wang et.al describes the role of Membrane Associated RNA- containing Vesicles (MARVs) in regulating the embryonic cell division. They particularly indicate in this study that MARVs containing DDX1 an activator of maternal RNA and Ca²⁺ localize with energized mitochondria at the subplasmalemmal region and drive embryonic cells to divide from 2 celled to multicell stage. Although, overall, it is an interesting study, the paper has several robustness and interpretation issues, which make these findings more speculative. Following are my comments for the manuscript.

- First, the manuscript is not written in a form which brings clarity to the story. Many of the figures are placed in sequence which make it hard to understand the message presented in the manuscript.
- The concept of J aggregates, which I think should have been described in greater detail in manuscript and presented in extended data Figure 1c is not described in the text. Moreover, it is difficult to understand what exactly extended Figure 1c is depicting.
- It is unclear from the Figure 1b, what is the positioning of the MARVs with respect to the plasma membrane in the 2 celled embryo stage. Also, the figure does not provide any quantification of this data.
- I think its quite important to have quantification of the data from extended data Figure 2, just by looking at the one representative electron micrograph, it is difficult to extrapolate too much about the data.
- It is difficult to follow the argument provided by the authors in Figure2b, where authors describe addition of 0.1% saponin causes a leakage of Ca²⁺ from various intracellular stores. It is unclear from this experiment how authors could specifically delineate loss of ring of Ca²⁺ ions from the subplasmalemmal region be affected by MERVs. A loss of plasma membrane itself from the 2 celled embryos could itself drive impaired distribution of Ca²⁺ in this situation. Authors should apply similar strategy as described in the references to depolarize the mitochondria or affect local calcium concentration by inhibiting plasma membrane calcium flux and thus decipher the role of MERVs in regulating local Ca²⁺ balance. Besides, the data described in extended data Figure 6 require further quantification to rule out the artifact effect from the data.
- Data in Figure3a is unclear regarding the genotype of the embryos that are stalled at 2-celled stage and those that reach 16-celled stage. It would be important to clearly delineate that the cause of dysregulated plasmalemmal Ca²⁺ distribution is due to DDX^{-/-} knock out. Moreover, authors indicate that at 16-celled embryo stage the Ca²⁺ redistributes back to subplasmalemmal region. Although, interesting but requires more in-depth analysis to determine the reason for such Ca²⁺ dynamics changes.

- The positioning of the mitochondrial membrane potential data depicted in Figure 3b should be clearer. It is confusing to see the representative images in the extended data Figure 7a and its quantification in Figure 3b.
- The authors indicate in the Figure 3d that compared to the wild type embryos the stalled embryos have higher mitochondrial ROS production. An elevated mitochondrial ROS production could occur due to several reasons, the authors should measure other parameters of mitochondrial energetics including ATP levels in the well genotyped embryos.
- It would be important to quantify the interaction of mitochondria and ER with subplasmalemmal Ca^{2+} .
- In Figure 4 quantification of colocalization is necessary.
- The model described in Figure 4b is bit exaggerated, the data in all figures regarding Ca^{2+} (Fluo4) is basically a snapshot and does not provide any kinetic information. Also, it is bit curious that Ca^{2+} can stay for long time in the local subplasmalemmal environment, without it being getting defused in the global Ca^{2+} pool in the cytosol.
- The manuscript is very speculative regarding the role of Ca^{2+} in MARV for embryogenesis, without much supporting evidence for the same.

Reviewer #2 (Remarks to the Author):

Godbout and co-workers report the discovery of an aggregation of a new cellular organelle that emerges in early stage embryos. The organelle contains calcium and the DEAD-box protein DDX1, which is required for the formation of the cellular structure. The authors present evidence that the organelle is associated with the regulation of the calcium distribution in embryonic cells.

The discovery of a new cellular organelle appears noteworthy. The finding that the organelle is linked to embryonic viability enhances the significance of the observation.

The paper relies exclusively on microscopy, which is certainly appropriate for the main topic of the manuscript, the description of the cellular structure. However, the authors also suggest that the structures play a critical role in the activation of the cytoplasmic polyadenylation machinery, based on co-localization of polyadenylation factors and DDX1 in the new organelle. This is a huge over-interpretation of the data shown. Co-localization does by no means imply any function. Without specific functional data backing the author's claims, this notion needs to be removed from the paper, even as hypothetical claim. This claim is misleading as presented, which is fairly prominent in abstract, text and even in figures, and thus a disservice for readers not intimately familiar with RNA metabolism.

Reviewer #3 (Remarks to the Author):

Wang and colleagues describe the characterization of DDX1 aggregates in pre-compaction embryos, identifying a ring-like structure containing RNA, Ca²⁺, DDX1 and other cytoplasmic polyadenylation factors which they term MARVs. Disruption of Ddx1 perturbs Ca²⁺ distribution and mitochondrial membrane potential, induces fragmentation, and as described previously, arrests development around the time of embryonic genome activation.

The study is interesting, extends their previous observations on DDX1 aggregates, and proposes a role for Ddx1 in cytoplasmic polyadenylation. I have several questions that need to be addressed to clarify their conclusions.

Specific comments

1. Given that Ddx1^{-/-} embryos arrest at the 2-4 cells stage, which coincides with embryonic genome activation requiring cytoplasmic polyadenylation to progressively activate maternal RNA stores, it would be important to understand some of the transcriptional changes that are affected. Insufficient activation of the embryonic genome alone will arrest development (in the absence of changes to Ca²⁺ distribution, and vice versa). Have the authors considered the role calcium plays in regulating CPEB phosphorylation?
2. It is difficult to see any relationship/co-localization of phosphorus and nitrogen (Fig 1) with Ddx1, and certainly not as depicted in 1e. At a minimum, the magnified section in Fig 1d needs to be replicated for the zero-loss phosphorus image, and magnified regions of the nitrogen images provided. The zero-loss nitrogen image seems particularly poor.
3. Line 59/Extended data Fig 2: The authors describe changing Ddx1 patterns through mouse embryo development. What cell types (trophectoderm or inner cell mass) were analyzed, and which is featured in the figure? Similarly, for the day E3.5 embryo, early lineage specification would have been initiated; are there differences in the appearance of MARVs across these two cell types? 'Late blastocyst' is also denoted – please indicate the embryonic day this equates to (E4.5?). This is important as both populations have two very different metabolic strategies.
4. Further, how do the authors reconcile the different Ddx1 aggregate localization patterns observed in the present study relative to their 2019 Dev Biol paper where aggregates are quite discernable at all stages? There seems to be particular disparity between what is presented in Figure 2a and 4a relative to this previous study (both are confocal).
5. The authors note that Ca²⁺ displayed a similar distribution pattern to Ddx1 aggregates. Is there any evidence of co-localization? The authors would need to perform FRET to resolve this.
6. Is Ddx1 also disrupted upon FCCP treatment?
7. Saponin is used to permeabilize the vesicle membrane, however the authors do not detect a loss of membrane integrity. Have the authors used another membrane marker to confirm this structure within

MARVs? As dashed lines are used to denote a MARV, it is virtually impossible to visualize any membrane surrounding the structure as indicated.

8. The stalled 2-cell embryo in Figure 3a appears more like an oocyte that has not correctly extruded its 2nd polar body or undergone cleavage leading to disproportionate cytoplasmic volume. As calcium is required for cleavage divisions, could this relate to altered Ca^{2+} regulation following fertilization that then contributes to disorganized Ddx1? i.e. are Ca^{2+} waves normal following fertilization in Ddx1 embryos? Is there any evidence to suggest that MARV membranes contain Ca^{2+} transporters? How frequent are these types of 'embryo' observed?

9. Ideally, dual staining each of embryo with Ddx1 is required (Figure 3). Similarly, dual staining of DAPI and mitotracker within the same embryo, and mtROS and mitotracker combined, would indicate clear relationships compared with individual staining on a small number of embryos. The representative mtROS image of the stalled embryo (Fig 3d) appears to indicate that fragmented cells and the polar bodies stain more intensely, while the remaining blastomere appears equivalent to the WT. For this reason, mean intensity should be determined per cell. What level of apoptosis is evident in Ddx1^{-/-} 2-cell embryos? Examination of this is particularly relevant given the suggestion that fragmentation is apparent with Ddx1 disruption.

Note that the stalled embryo in 3d is likely a fragmented 2-cell, though would be hard to discern without timelapse imaging.

10. The distributed mitochondria in Figure 5 do not correlate with the observed pattern of punctate mitochondrial localization shown in Fig 3c. The authors suggest that the altered mitochondrial membrane potential might indicate an increased need for ATP. Are there differences in ATP levels in Ddx1^{-/-} embryos?

11. There seem to be several Ddx1 aggregates that do not stain for CPEB1, and likewise many areas of staining for CPSF2 that do not overlay with Ddx1 staining. For the former, could these represent sites where polyadenylation is not very active? And for the latter, what structures might it be localizing to (if not the nucleus). Is localization of CPSF2 and CPEB1 lost in Ddx1^{-/-} 2-cell embryos?

Minor comments:

Line 107: please add a statement that the FCCP data are not shown.

Line 198/205: do the authors mean 300 IU/ml hyaluronidase (not ug/ml)?

Line 203: why were mice primed to obtain MII oocytes here, when all other aspects were conducted on naturally mated mice?

M16 medium is a very deficient culture option that does not mimic the in vivo environment sufficiently. While I understand it is commercially available, it does not contain components that are important for appropriately supporting embryo development beyond the 2-cell stage, particularly amino acids. Indeed, CPEB is involved in the cellular response to amino acids. Similarly, exposure to medium lacking amino acids would alter ATP generating pathways. Given that Ddx1 is ATP-dependent, could the results here in

part relate to insufficient nutrient support (combined with a potentially more susceptible/sensitive embryo)?

Figure 3: please clarify that 16-cell embryos are from WT in the legend and figure.

Figure 4b: the legend for this model states that MARVs form upon fertilization, but this (syngamy) has not been examined.

Figure 5: include the localization of mitochondria for 'wildtype'

We thank all three reviewers for their comments and suggestions and hope that our manuscript has been substantially improved by the additional experiments, clarifications and quantitation data.

Reviewer #1 (Remarks to the Author):

The paper “DDX1 Vesicles Control Calcium-dependent Mitochondrial Activity in Embryos” by Wang et.al describes the role of Membrane Associated RNA-containing Vesicles (MARVs) in regulating the embryonic cell division. They particularly indicate in this study that MARVs containing DDX1 an activator of maternal RNA and Ca²⁺ localize with energized mitochondria at the subplasmalemmal region and drive embryonic cells to divide from 2 celled to multicell stage. Although, overall, it is an interesting study, the paper has several robustness and interpretation issues, which make these findings more speculative. Following are my comments for the manuscript.

• First, the manuscript is not written in a form which brings clarity to the story. Many of the figures are placed in sequence which make it hard to understand the message presented in the manuscript.

The manuscript has been extensively revised. We have combined the original Figure 2a with Extended Data Figure 5, and Figures 2b and c with Extended Data Figure 6. We have also included mitochondria fragmentation quantification, JC-1 quantification data in Figure 5 (original Figure 3) along with images. We have included additional data (ATP levels, FCCP treatment etc., see below) and have incorporated additional panels to figures where requested or appropriate. We now provide all requested quantification data. Hopefully, the new data and order of figures will help make the manuscript easier to read with a clearer message.

• The concept of J aggregates, which I think should have been described in greater detail in manuscript and presented in extended data Figure 1c is not described in the text. Moreover, it is difficult to understand what exactly extended Figure 1c is depicting.

We now describe the concept of J aggregates on pg. 3, lines 44-54. In the Extended Data Figure 1c, we show the staining of a wild-type 2-cell embryo with JC-1 with the monomeric form (green) mostly found in the inner cytoplasm and the aggregate form (magenta) mostly located in the subplasmalemmal space.

• It is unclear from the Figure 1b, what is the positioning of the MARVs with respect to the plasma membrane in the 2 celled embryo stage. Also, the figure does not provide any quantification of this data.

We now provide the requested quantification data of the positioning of MARVs in 2-cell stage embryos on pg. 4, lines 60-61. The average distance of MARVs to the plasma membrane is $1.46 \pm 0.55 \mu\text{m}$.

• I think its quite important to have quantification of the data from extended data Figure 2, just by looking at the one representative electron micrograph, it is difficult to extrapolate too much about the data.

We now quantitate amounts of DDX1 in membrane-bound vesicles. At E0.5, when MARVs first appear, ~40% of DDX1 is in membrane-bound vesicles. In E1.5 to E3.5 embryos, ~83% of DDX1 if found in membrane-bound vesicles (pg. 4, lines 66-69).

Since DDX1 is mostly found in membrane-bound vesicles at E1.5 – 3.5, the sizes of the DDX1 aggregates as previously published (2019 Developmental Biology ref 9) likely reflect the sizes of MARVs. Quantification of MARVs using TEM is going to be much less accurate as we don't have the technology to generate TEM stacks.

• It is difficult to follow the argument provided by the authors in Figure2b, where authors describe addition of 0.1% saponin causes a leakage of Ca²⁺ from various intracellular stores. It is unclear from this experiment how authors could specifically delineate loss of ring of Ca²⁺ ions from the subplasmalemmal region be affected by MARVs. A loss of plasma membrane itself from the 2 celled embryos could itself drive impaired distribution of Ca²⁺ in this situation.

Authors should apply similar strategy as described in the references to depolarize the mitochondria or affect local calcium concentration by inhibiting plasma membrane calcium flux and thus decipher the role of MARVs in regulating local Ca²⁺ balance. Besides, the data described in extended data Figure 6 require further quantification to rule out the artifact effect from the data.

The purpose of the experiment was to examine the location of Ca²⁺ upon disruption of the vesicle membranes in MARVs. However, as indicated by the reviewer, saponin will permeabilize the plasma membrane as well as organelles. Our results indicate that 0.1% saponin treatment increases accumulation of Ca²⁺ in mitochondria based on co-staining with Fluo-4 AM and MitoTracker. Although indirect, and as there is so much Ca²⁺ in MARVs, these results suggest that when Ca²⁺ is released from MARVs, a natural route of Ca²⁺ redistribution may be mitochondria.

We used two different approaches to address the reviewer's questions regarding the role of MARVs in regulating local Ca²⁺ balance. First, as recommended by the reviewer and previously described in the literature, we treated 2-cell embryos with FCCP which depolarizes mitochondria membrane potential and disrupts Ca²⁺ microdomains in 2-cell embryos. After verifying that FCCP treatment resulted in disappearance of Ca²⁺ microdomains as previously reported in the literature (Manser et al. J Cell Sci 2006, Nagaraj et al. Cell 2017), we focused on quantifying the effect of FCCP treatment on MARVs. We found that there was a significant increase in small DDX1 aggregates of <0.5 μm³ in size compared to DMSO control (pgs. 6-7; Figure 3b).

In a further attempt to address the role of MARVs in regulating local Ca^{2+} distribution, we examined DDX1 aggregates in embryos cultured under Ca^{2+} -free conditions (pgs. 6-7). First, 2-cell embryos were cultured for up to 48 h in Ca^{2+} free medium. As previously reported, we found that embryos cultured under Ca^{2+} -free conditions were able to develop to the 8- to 16-cell stage after 48 h in culture but failed to compact or reach the blastocyst stage (Extended Data Figure 5e). Second, we studied the levels and distribution of Ca^{2+} in 2-cell embryos cultured under Ca^{2+} -free conditions for 24 h. We observed an overall ~50% decrease in Ca^{2+} levels compared to embryos cultured in regular medium (Extended Data Figure 5b). These embryos divided normally after 24 h culture and remain metabolically active (Extended Data Figure 5c; 5d). In spite of this decrease in Ca^{2+} levels, Ca^{2+} aggregates were still clearly present. MARVs were then quantitated based on size. Similar to FCCP treatment, we found a significant increase in small DDX1 aggregates of $<0.5 \mu\text{m}^3$ in size when cultured in Ca^{2+} -free condition compared to regular medium (Figure 3d). We also observed a significant decrease in the frequency of the largest MARVs ($>5 \mu\text{m}^3$) (Figure 3d). Moreover, we found a ~75% increase in the average number of DDX1 aggregates and a ~34% overall increase in average DDX1 cluster size compared to embryos cultured in regular medium (pgs. 6-7, lines 137-140 and Figure 3e).

Although none of the experiments demonstrate how MARVs regulate local Ca^{2+} balance, when taken in combination, these different approaches suggest a functional link between MARVs and Ca^{2+} levels/ Ca^{2+} spatial distribution in embryos.

• Data in Figure 3a is unclear regarding the genotype of the embryos that are stalled at 2-celled stage and those that reach 16-celled stage. It would be important to clearly delineate that the cause of dysregulated plasmalemmal Ca^{2+} distribution is due to DDX^{-/-} knock out.

We agree with the reviewer that stalled 2-cell embryos should be genotyped to verify that they are *Ddx1^{-/-}*. However, repeated attempts to carry out genotype analysis on stalled embryos have not been successful. We believe that this is due to the nuclear fragmentation that accompanies embryo stalling. However, we have carried out extensive genotype analysis of embryos from heterozygote crosses taken directly from the mother. Genotype analysis indicate that ~one quarter of embryos from such crosses are *Ddx1^{-/-}* (Godbout lab unpublished data; please see attached image below). Albeit indirect, our embryonic culture work is in agreement with these data, indicating that ~one quarter of embryos from heterozygote crosses stall in culture at the 2 to 4 cell stage. This is now discussed on pg. 8, lines 164-168). These data are in agreement with our previous publication (Developmental Biology 2019)

Moreover, authors indicate that at 16-celled embryo stage the Ca²⁺ redistributes back to subplasmalemmal region. Although, interesting but requires more in-depth analysis to determine the reason for such Ca²⁺ dynamics changes

The 16-cell embryo originally shown in Figure 3 does show Ca²⁺ aggregates, but these are not located in the subplasmalemmal space. To further address Ca²⁺ dynamics, we have now carried out Ca²⁺ distribution analysis in embryos from 2-cell to blastocyst stages (pgs. 8-9, lines 170-175 and Figure 5a). We found that the distribution of Ca²⁺ changes from a primarily subplasmalemmal cytoplasmic distribution at the 2- to 4-cell stages to a more inner cytoplasmic distribution after compaction stages. We did not observe Ca²⁺ redistribution to the subplasmalemmal region.

We postulate that the distribution of Ca²⁺ and MARVs in 2 to 4-cell stage embryos is likely to be affected not only by Ca²⁺ but also by active mitochondria. During compaction of later-stage embryos, Ca²⁺ is no longer found in the subplasmalemmal cytoplasm where the active mitochondria reside. This may reflect changes in metabolism and mitochondria state in preparation for entry into the blastocyst stage.

• The positioning of the mitochondrial membrane potential data depicted in Figure3b should be clearer. It is confusing to see the representative images in the extended data Figure7a and its quantification in Figure3b.

We have corrected this in our revised manuscript and have now placed the quantification data alongside Figure 5b.

• The authors indicate in the Figure3d that compared to the wild type embryos the stalled embryos have higher mitochondrial ROS production. An elevated mitochondrial ROS production could occur due to several reasons, the authors should measure other parameters of mitochondrial energetics including ATP levels in the well genotyped embryos.

We have now measured ATP levels in non-stalled and stalled embryos. As indicated above, we are not able to directly genotype stalled embryos; however, genotype analysis of embryos from heterozygote crosses indicate that ~one-quarter of embryos from such crosses are *Ddx1*^{-/-} which is aligned with the number of stalled embryos. Our

results indicate that the stalled embryos have more ATP compared to non-stalled embryos (pgs. 9, lines 180-184 and Figures 6a; 6b).

• It would be important to quantify the interaction of mitochondria and ER with subplasmalemmal Ca²⁺.

We have added the requested quantification using the Pearson's Correlation Coefficient. Our data indicate that ER and mitochondria do not co-compartmentalize with subplasmalemmal Ca²⁺ (Figure 2e, pg. 6).

• In Figure 4 quantification of colocalization is necessary.

We have added the requested quantification data (see pg. 10, lines 212-214 and Extended Data Figure 7).

• The model described in Figure 4b is bit exaggerated, the data in all figures regarding Ca²⁺ (Fluo4) is basically a snapshot and does not provide any kinetic information. Also, it is bit curious that Ca²⁺ can stay for long time in the local subplasmalemmal environment, without it being getting defused in the global Ca²⁺ pool in the cytosol.

Our data indicate that Ca²⁺ remains in the subplasmalemmal environment from the 2-cell to 4-cell stages. To further address this issue, we compared Ca²⁺ subcellular localization in 2-cell embryos cultured in Ca²⁺ free media and Ca²⁺ containing media for 24 h. Intriguingly, Ca²⁺ remained in the subplasmalemmal environment even when embryos were cultured under Ca²⁺ free condition (pg. 7 lines 134-137 and Extended Data Figure 5a; 5b).

Although not included in the revised manuscript, we also attempted to study Ca²⁺ distribution in 2-cell embryos by carrying out time-lapse live cell imaging on embryos cultured in the presence of Fluo-4 AM dye. Although Fluo-4 AM was not directly toxic to the embryo, photobleaching and phototoxicity was observed, precluding analysis beyond 2-4 h.

• The manuscript is very speculative regarding the role of Ca²⁺ in MARV for embryogenesis, without much supporting evidence for the same.

Hopefully, the additional experiments and quantification data requested by the reviewer will strengthen the manuscript. The discovery of novel membrane-bound organelles clustered to form aggregates that are rich in Ca²⁺, along with the demonstrated effects of DDX1 knockout on MARVs, Ca²⁺ distribution and mitochondria, point to MARVs as important structures in the control of early-stage embryonic development. We postulate that Ca²⁺ localization in MARVs enables more precise control of Ca²⁺ release which is critical for energy production in embryonic development (please see Supplementary Discussion).

Reviewer #2 (Remarks to the Author):

Godbout and co-workers report the discovery of an aggregation of a new cellular organelle that emerges in early-stage embryos. The organelle contains calcium and the DEAD-box protein DDX1, which is required for the formation of the cellular structure. The authors present evidence that the organelle is associated with the regulation of the calcium distribution in embryonic cells.

The discovery of a new cellular organelle appears noteworthy. The finding that the organelle is linked to embryonic viability enhances the significance of the observation.

The paper relies exclusively on microscopy, which is certainly appropriate for the main topic of the manuscript, the description of the cellular structure. However, the authors also suggest that the structures play a critical role in the activation of the cytoplasmic polyadenylation machinery, based on co-localization of polyadenylation factors and DDX1 in the new organelle. This is a huge over-interpretation of the data shown. Co-localization does by no means imply any function. Without specific functional data backing the author's claims, this notion needs to be removed from the paper, even as hypothetical claim. This claim is misleading as presented, which is fairly prominent in abstract, text and even in figures, and thus a disservice for readers no intimately familiar with RNA metabolism.

We now simply indicate that cytoplasmic polyadenylation factors CPEB1 and CPSF2 localize to MARVs. We have moved these data to Extended Data Figure 7. Any reference to a role for MARVs in activation of the cytoplasmic polyadenylation machinery has been removed from the manuscript. We have also removed the cytoplasmic polyadenylation model presented in the previous version of the manuscript.

Reviewer #3 (Remarks to the Author):

Wang and colleagues describe the characterization of DDX1 aggregates in pre-compaction embryos, identifying a ring-like structure containing RNA, Ca²⁺, DDX1 and other cytoplasmic polyadenylation factors which they term MARVs. Disruption of Ddx1 perturbs Ca²⁺ distribution and mitochondrial membrane potential, induces fragmentation, and as described previously, arrests development around the time of embryonic genome activation.

The study is interesting, extends their previous observations on DDX1 aggregates, and proposes a role for Ddx1 in cytoplasmic polyadenylation. I have several questions that need to be addressed to clarify their conclusions.

Specific comments

1. Given that Ddx1^{-/-} embryos arrest at the 2-4 cells stage, which coincides with

embryonic genome activation requiring cytoplasmic polyadenylation to progressively activate maternal RNA stores, it would be important to understand some of the transcriptional changes that are affected. Insufficient activation of the embryonic genome alone will arrest development (in the absence of changes to Ca²⁺ distribution, and vice versa). Have the authors considered the role calcium plays in regulating CPEB phosphorylation?

To better understand the changes in mRNA transcripts resulting from *Ddx1* knockout in early-stage embryos, we carried out single embryo RNA sequencing of 18 embryos from 2 heterozygote crosses. As detailed in the revised manuscript, we found ~300 genes and 90 genes that were downregulated and upregulated, respectively, in embryos with low *Ddx1* RNA (presumed *Ddx1*^{-/-}) versus high *Ddx1* RNA (presumed wild-type or *Ddx1* heterozygotes) (pg. 10, lines 197-204 Extended Data Table1). Some of these genes are associated with mitochondrial functions (e.g. *Commd4*, *Ucp2*, *Top1mt*, *Ndufaf1*); Ca²⁺ (e.g. *Cacna1*, *Calcoco1*, *Slc24a3*), and ROS production (e.g. *Cox19*, *Top1mt*, *Ucp2*).

Yes, we are indeed aware and very interested in the possibility that Ca²⁺ may be involved in regulating CPEB1 phosphorylation. Unfortunately, in spite of several requests, we were unable to obtain an antibody that specifically recognizes phosphorylated CPEB protein to pursue this investigation. As per Reviewer 2's comments, due to the lack of hard evidence, we have removed our claim of a possible role for MARVs in cytoplasmic polyadenylation.

2. It is difficult to see any relationship/co-localization of phosphorus and nitrogen (Fig 1) with Ddx1, and certainly not as depicted in 1e. At a minimum, the magnified section in Fig 1d needs to be to be replicated for the zero-loss phosphorus image, and magnified regions of the nitrogen images provided. The zero-loss nitrogen image seems particularly poor.

The main purpose of the EFTEM images shown in Figure 1 was to demonstrate that MARVs contain RNA (phosphorus) and protein (nitrogen). We have tried doing phosphorus EFTEM with DDX1 immunogold labelling. However, although DDX1 could be detected, we lost the phosphorus signal. We believe that this may be the result of RNA degradation caused by 3 days of processing time. In the revised manuscript, we now show that >80% of DDX1 is found in MARVs (see Reviewer 1 comment). In the zero-loss phosphorus image, we are able to identify the membrane bound vesicles found in MARVs as the protocol for phosphorus detection is compatible with staining the membranes in vesicles with OsO₄. Since we know that the great majority of DDX1 is in membrane-bound vesicles, and the phosphorus localizes to the membrane bound vesicles, we conclude that phosphorus is found in DDX1-containing membrane-bound vesicles.

We now provide the requested magnified image of nitrogen and DDX1 immunogold (Figure 1).

As far as the zero-loss nitrogen image, we now include better images (Figure 1) and apologize for the poor-quality image provided in the original manuscript.

3. Line 59/Extended data Fig 2: The authors describe changing Ddx1 patterns through mouse embryo development. What cell types (trophectoderm or inner cell mass) were analyzed, and which is featured in the figure? Similarly, for the day E3.5 embryo, early lineage specification would have been initiated; are there differences in the appearance of MARVs across these two cell types? ‘Late blastocyst’ is also denoted – please indicate the embryonic day this equates to (E4.5?). This is important as both populations have two very different metabolic strategies.

In the Figure 4 of our 2019 Dev Biol paper, we showed that the number of cytoplasmic DDX1 aggregates gradually decreased with embryonic development. When embryos are cultured until they hatch, levels of DDX1 in the nucleus gradually increase. However, we did not observe significant differences in DDX1 distribution in ICM versus trophoctoderm cells.

In the revised manuscript, we include images showing MARVs in the ICM and trophoctoderm cells of E3.5 embryos (Extended Data Figure 2b). In agreement with our 2019 Dev Biol paper, we found no difference in the localization of MARVs or DDX1 localization in MARVs in ICM versus trophoctoderm.

Yes, “late blastocyst” is around E4.5 (now indicated in Extended Figure 2). E1.5 embryos were cultured for 72 h to reach the late blastocyst stage.

4. Further, how do the authors reconcile the different Ddx1 aggregate localization patterns observed in the present study relative to their 2019 Dev Biol paper where aggregates are quite discernable at all stages? There seems to be particular disparity between what is presented in Figure 2a and 4a relative to this previous study (both are confocal).

We apologize for the confusion. In our 2019 Developmental Biology paper, we presented the images as maximum intensity Z-stack projections because we were focusing on the changes in sizes and numbers of the DDX1 aggregates. In the current manuscript, we used single plane rather than Z-stack projections for clearer depiction of MARV distribution. Maximum intensity Z-stack images of DDX1-immunostained embryos in Figure 3 and Extended Data Figure 4 show similar distribution of DDX1 aggregates as we previously published.

Also, please note that we used directly-conjugated DDX1 antibody in the current manuscript. The directly-conjugated antibody reveals clear differences in the shape of the DDX1 aggregates (granular vs ring-like) even by confocal microscopy.

5. The authors note that Ca²⁺ displayed a similar distribution pattern to Ddx1 aggregates. Is there any evidence of co-localization? The authors would need to perform FRET to resolve this.

The reviewer is correct in that our experiments don't show whether Ca²⁺ and DDX1 co-localize in MARVs. We've inquired about doing this experiment; however, FRET using Fluo-4 AM and anti-DDX1 antibody don't appear to be technically feasible, at least with the instruments that we have. The Fluo-4 AM dye is used on live cells whereas our anti-DDX1 antibody requires permeabilization and fixation. The latter processes result in a considerably weakened Fluo-4 AM signal, precluding any meaningful FRET analysis.

As we were primarily interested in whether Ca²⁺ and DDX1 co-compartmentalize in MARVs, we carried out Pearson's correlation analysis of Ca²⁺ and DDX1, and found that Ca²⁺ and DDX1 do indeed co-compartmentalize. Quantification is now provided as per Reviewer 1's request (pg. 6; Figure 2e).

6. Is Ddx1 also disrupted upon FCCP treatment?

Upon FCCP treatment for 2 h, we observed a significant increase in small DDX1 aggregates of <0.5 μm³ (Figure 3b). More significant changes were observed when embryos were cultured under Ca²⁺-free condition (Figure 3d). Thus, the size of DDX1 aggregates is affected by mitochondrial membrane potential and Ca²⁺ distribution. Also please see Reviewer 1 comments.

7. Saponin is used to permeabilize the vesicle membrane, however the authors do not detect a loss of membrane integrity. Have the authors used another membrane marker to confirm this structure within MARVs? As dashed lines are used to denote a MARV, it is virtually impossible to visualize any membrane surrounding the structure as indicated.

Saponin is a mild detergent-like molecule used for live cell experiments to permeabilize cells by removing cholesterol in membranes. Saponin does not destroy membranes (Willingham et al. PNAS 75: 4359-4363, 1978). At this time, we have had to rely on electron microscopy for detection of membranes surrounding DDX1 vesicles. One of our future goals will be to characterize the membrane of DDX1 vesicles.

We have reduced the number of dashes to allow visualization of membrane-surrounded vesicles in MARVs (now Figure 4b).

8. The stalled 2-cell embryo in Figure 3a appears more like an oocyte that has not correctly extruded its 2nd polar body or undergone cleavage leading to disproportionate cytoplasmic volume. As calcium is required for cleavage divisions, could this relate to altered Ca²⁺ regulation following fertilization that then contributes to disorganized Ddx1? i.e. are Ca²⁺ waves normal following fertilization in Ddx1 embryos? Is there any evidence to suggest that MARV

membranes contain Ca²⁺ transporters? How frequent are these types of ‘embryo’ observed?

The stalled embryo in the previous Figure 3a is an actual 2-cell stage embryo positioned at an angle. The revised figure (Figure 5a) more clearly shows a stalled 2-cell embryo. Please note that we were not able to obtain clear images of Z-stacks of these embryos potentially due to photobleaching. A Z-stack would have provided a more comprehensive image of the embryos even when positioned at an angle.

As pointed out by the reviewer, Ca²⁺ dysregulation may affect the Ca²⁺ waves following fertilization. Based on our previous work (2019 Developmental Biology), approximately one-quarter of the embryos from heterozygote *Ddx1* crosses stall at the 2-cell stage with some embryos developing to 4 cells. These data suggest that stalling must occur later than the Ca²⁺ waves that follow fertilization.

As to the Ca²⁺ transporters, we have not yet investigated this particular aspect of the vesicles. We will need to figure out which Ca²⁺ transporters to focus on. This will be addressed in future studies.

9. Ideally, dual staining each of embryo with Ddx1 is required (Figure 3). Similarly, dual staining of DAPI and mitotracker within the same embryo, and mtROS and mitotracker combined, would indicate clear relationships compared with individual staining on a small number of embryos. The representative mtROS image of the stalled embryo (Fig 3d) appears to indicate that fragmented cells and the polar bodies stain more intensely, while the remaining blastomere appears equivalent to the WT. For this reason, mean intensity should be determined per cell. What level of apoptosis is evident in *Ddx1*^{-/-} 2-cell embryos? Examination of this is particularly relevant given the suggestion that fragmentation is apparent with *Ddx1* disruption.

Note that the stalled embryo in 3d is likely a fragmented 2-cell, though would be hard to discern without timelapse imaging.

DDX1 immunostaining is not compatible with staining with live cell dyes such MitoSOX and JC-1. DDX1 immunostaining requires permeabilization and fixation, processes that are not compatible with live cell dye staining.

In an attempt to address the reviewer’s request for dual staining, we now include triple staining data with Hoechst (nucleus), MitoSOX (mtROS) and MitoTracker Deep Red (mitochondria) in Figure 6 with weighted average index construction of these three channels to better address the relationship between nuclear fragmentation, mitochondrial ROS, and mitochondrial fragmentation. We also include quantification of the original individual staining data in Figure 5 as there are drawbacks to quantifying data on images with multiple fluorophores, as photons from one channel may be collected by other channels and lead to inaccurate quantification.

As far as mean intensity per cell, all our live-cell images were taken using ultra-thin bottom 96 well plates with a 20X lens. As a result, although similar slice numbers are obtained for different embryos, the embryos may be positioned such that the laser intensity is not the same throughout the embryo (e.g. with the embryo being positioned at an angle whereby one side is higher than the other). This may result in the dimmer appearance of the cells further away from the imaging lens. However, as we are measuring mean intensity levels with ROI drawn, no significant difference in mean intensity values between embryos positioned in different angle are observed (please refer to the error bars on Figure 5h and Figure 6d). Moreover, given that we have analyzed 19 stalled embryos in total (7 in Figure 5h and 12 in Figure 6d), any effect of different positioning is likely to even out. We have replaced the image used in the previous version of the manuscript with an image (Figure 5g) of a better positioned stalled embryo. Hopefully, the new figure better illustrates signal intensities. We measured mean intensity per embryo rather than mean intensity per cell because at the 2-cell stage, apoptosis of one cell will affect the development of the entire embryo. Measuring mean intensity per cell may be misleading because of slight positional differences between the 2 cells in the embryo.

Unfortunately, our attempts to carry out timelapse imaging were unsuccessful because of photobleaching and cytotoxicity of MitoSOX.

10. The distributed mitochondria in Figure 5 do not correlate with the observed pattern of punctate mitochondrial localization shown in Fig 3c. The authors suggest that the altered mitochondrial membrane potential might indicate an increased need for ATP. Are there differences in ATP levels in Ddx1^{-/-} embryos?

We have revised Figure 5 (now Figure 7) so that it is more representative.

As recommended by the reviewer, we now include quantification of ATP levels in wild-type versus stalled embryos. Stalled embryos have higher levels of ATP compared to wild-type 2-cell embryos (pgs. 8-9, lines 180-184 and Figures 6a; 6b).

11. There seems to be several Ddx1 aggregates that do not stain for CPEB1, and likewise many areas of staining for CPSF2 that do not overlay with Ddx1 staining. For the former, could these represent sites where polyadenylation is not very active? And for the latter, what structures might it be localizing to (if not the nucleus). Is localization of CPSF2 and CPEB1 lost in Ddx1^{-/-} 2-cell embryos?

This is a very interesting question. It will be important to figure out whether DDX1 aggregates that do not stain for CPEB1 are less active sites of polyadenylation. We tried very hard to obtain an antibody to phosphorylated CPEB1 to address this question, but without success. Also, please refer to our response to the first comment of this reviewer and Reviewer 2 comments.

We did investigate the localization of CPEB1 and CPSF2 in stalled embryos (please see image below). The Pearson's Correlation Coefficient with regards to co-

compartmentalization with DDX1 is higher for both CPEB1 and CPSF2 in the stalled embryos compared to wild-type 2-cell embryos. Considering that stalled embryos show considerably reduced nuclear staining of both CPEB1 and CPSF2, it may not be surprising to see an increase in the Pearson's Correlation Coefficient. These data suggest that polyadenylation factors CPEB1 and CPSF2 are retained in residual MARVs of stalled embryos. However, given the preliminary work of this aspect of our work (see Reviewer 2 comment), we only show CPEB1 and CPSF2 immunostaining data in extended data Figure 7 and have removed conjectures about a possible role for MARVs in cytoplasmic polyadenylation from the manuscript. We now indicate that this will be the subject of future investigations.

Minor comments:

Line 107: please add a statement that the FCCP data are not shown.

We now indicate that the FCCP data are not shown.

Line 198/205: do the authors mean 300 IU/ml hyaluronidase (not ug/ml)?

According to the manufacturer (H4272, Millipore Sigma), the hyaluronidase we purchased is suitable for mouse embryo cell culture and has the specific activity of 750-3000 unit/mg solid. The recommended usage is ~300 µg/ml. This information is now included on pg. 12, lines 246-248.

Line 203: why were mice primed to obtain MII oocytes here, when all other aspects were conducted on naturally mated mice?

We try to avoid superovulation whenever possible as superovulation has been shown to affect mitochondrial membrane potential and embryonic development (Shu et al. 2016 *Reproduction, Fertility and Development*. DOI: 10.1071/RD14300; Komatsu et al. 2014 *Reproduction*. DOI: 10.1530/REP-13-0288). However, as production of MII oocytes is dependent on the mouse estrogen cycle, it is difficult to carry out work on MII stage oocytes without superovulation. As we were not studying metabolism in MII stage oocytes, superovulation should not affect our results.

M16 medium is a very deficient culture option that does not mimic the in vivo environment sufficiently. While I understand it is commercially available, it does not contain components that are important for appropriately supporting embryo development beyond the 2-cell stage, particularly amino acids. Indeed, CPEB is involved in the cellular response to amino acids. Similarly, exposure to medium lacking amino acids would alter ATP generating pathways. Given that Ddx1 is ATP-dependent, could the results here in part relate to insufficient nutrient support (combined with a potentially more susceptible/sensitive embryo)?

To ensure that the stalled embryos are not caused by culturing embryos in M16 medium, we carried out experiments with embryos from heterozygote crosses cultured in the much more nutrient-rich KSOM medium. We tested embryos from 3 different heterozygous crosses (a total of 21 embryos). We identified 5 stalled embryos from these three crosses, suggesting that culture media is not a major factor in generating stalled embryos.

Figure 3: please clarify that 16-cell embryos are from WT in the legend and figure.

We now specify that the range of embryos stained with Fluo-4 AM in Fig. 5a are from cultured embryos from wild-type crosses (plated at the 1-cell stage) (pg. 8-9, line 170-175).

Figure 4b: the legend for this model states that MARVs form upon fertilization, but this (syngamy) has not been examined.

The point is well taken as MARVs form at 1-cell stage. This model figure has been deleted from the manuscript (see Reviewer 2's comments).

Figure 5: include the localization of mitochondria for 'wildtype'

We now include mitochondria for wildtype in Figure 7.

REVIEWER COMMENTS

Reviewer #1 (Remarks to the Author):

The authors have addressed all comments that were raised by the reviewer previously. The revised manuscript is suitable for publication in Nature Communications.

Reviewer #2 (Remarks to the Author):

The authors have responded constructively to the previous comments. No further issues.

Reviewer #3 (Remarks to the Author):

The authors have addressed several of the main points raised, with the edits and additional analyses improving the manuscript and associated conclusions.

Some minor points that need to be addressed:

The authors use FCCP, a mitochondrial protonophore that disrupts ATP synthesis, as a means to modulate Ca²⁺. This is an indirect way of modulating Ca²⁺ which does not eliminate the potential that metabolic signaling from mitochondria might be involved in MARV and Ca²⁺ localisation. A more appropriate modulator of Ca²⁺ localisation should be used (e.g. EGTA-AM).

In examining Ddx aggregate size and JC-1 staining in embryos grown in Ca²⁺ free medium, the authors suggest that the presence of high membrane potential indicates that these embryos are 'metabolically active' (which implies that embryos must be capable of continued development). However, to their surprise they later note that arrested embryos have high levels of 'active' JC-1 staining. This is in fact not surprising given that embryos that fail to develop display increased ATP synthesis, presumably in an attempt to keep up with the metabolic demands of development. The authors should clarify what they mean by 'metabolically active' and its implications, and should also be explicit in the main text as to what embryo stage was assessed.

Likewise, I also question why stalled embryos were only imaged after 72h in culture? I wonder whether this is too late to identify whether Ca²⁺ microdomains were present at any point. Does this mean that stalled 2-cell embryos were compared with later stage 'control' embryos (e.g. 4-cell as depicted in Fig 7).

This is effectively a comparison of apples and oranges. Embryos visualised upon collection (or at first cleavage), then cultured individually such that they could be retrospectively categorised, would ensure staging and timing is comparable.

The added RNAseq data is of interest, however an n=2 'Ddx low' embryos is not ideal for analysis. Can the authors confirm whether independent biological samples were used in the validation of the DEGs by qPCR? Further, the authors should detail how GAPDH was selected as the appropriate housekeeper, given its role in energy dynamics and Ca²⁺ signalling. The use of additional housekeepers, or a geometric mean would be more appropriate.

Have the authors tried isolating ICMs from blastocysts to examine Ca²⁺ localisation using Fluo-Am (given the trouble of uptake by the ICM in intact blastocysts using any molecular probe)?

In the conclusion, the authors focus on the potential role of MARVs in modulating mitochondrial function(via Ca²⁺), referencing stalled embryos, however as noted in my previous comments, developmental arrest may equally relate to inadequate activation of the embryonic genome which occurs around the 2-cell stage in mice. This is plausible given the RNAseq data obtained and potential role for MARVs in RNA processing. Based on how embryos have been compared, as noted above, I am not convinced of a cause+effect relationship for MARVs and timed Ca²⁺ release.

Please clarify that the mitochondrial number that is halved is 'per cell' (line 92).

Response to Reviewers' comments

Reviewers 1 and 2 were satisfied with our previous revisions

Reviewer #3 (Remarks to the Author):

The authors have addressed several of the main points raised, with the edits and additional analyses improving the manuscript and associated conclusions.

Some minor points that need to be addressed:

The authors use FCCP, a mitochondrial protonophore that disrupts ATP synthesis, as a means to modulate Ca²⁺. This is an indirect way of modulating Ca²⁺ which does not eliminate the potential that metabolic signaling from mitochondria might be involved in MARV and Ca²⁺ localisation. A more appropriate modulator of Ca²⁺ localisation should be used (e.g. EGTA-AM).

We have carried out the requested experiment and included the data for EGTA-AM in Figs. 4a-4d. Treatment with EGTA-AM for 3 hours resulted in an increase in small DDX1 aggregates (<0.5 μm^3) and a reduction of large DDX1 aggregates (>5 μm^3). Moreover, we now observe statistical significance for numbers of DDX1 aggregates >0.5 μm^3 , suggesting that EGTA-AM has a stronger effect on DDX1 aggregates than FCCP treatment.

In examining Ddx aggregate size and JC-1 staining in embryos grown in Ca²⁺ free medium, the authors suggest that the presence of high membrane potential indicates that these embryos are 'metabolically active' (which implies that embryos must be capable of continued development). However, to their surprise they later note that arrested embryos have high levels of 'active' JC-1 staining. This is in fact not surprising given that embryos that fail to develop display increased ATP synthesis, presumably in an attempt to keep up with the metabolic demands of development. The authors should clarify what they mean by 'metabolically active' and its implications, and should also be explicit in the main text as to what embryo stage was assessed.

The experiment where we stain the embryos with JC-1 was done in order to show that these embryos are alive and can undergo further development. Therefore, the DDX1 pattern change is not due to the lack of viability of the embryo. We have clarified this point in our main text (pg. 7; lines 144-146). We have also removed the 'intriguingly' where we show the JC-1 staining data. We found it surprising that the developmental failure of stalled embryos was associated with high levels of JC-1 staining. However, the reviewer is correct that our subsequent observations showing increased ATP synthesis in stalled embryos provides an explanation for the JC-1 staining data.

Likewise, I also question why stalled embryos were only imaged after 72h in culture? I wonder whether this is too late to identify whether Ca²⁺ microdomains were present at any point. Does this mean that stalled 2-cell embryos were compared with later stage 'control' embryos (e.g. 4-cell as depicted in Fig 7). This is effectively a comparison of apples and oranges. Embryos visualised upon collection (or at first cleavage), then cultured individually such that they could be retrospectively categorised, would ensure staging and timing is comparable.

We already know that we need to collect embryos at the 1-cell stage in order to identify stalled embryos in culture. To address the reviewer's question, we therefore collected embryos (from a total of 3 wild-type crosses, each cross analysed separately) at the 1-cell stage, cultured them to the 2-cell stage, treated the embryos with 5 μ M Fluo-4 AM, then divided the embryos into a 'control' group (N=11) and an 'imaging' group (N=10). All 2-cell embryos (N=10) imaged in culture failed to develop past the 2-cell stage because of phototoxicity. Embryos in the control group (N=11) which were not imaged developed normally to the blastocyst stage. Therefore, we were unable to perform the suggested experiment. We also tested embryos cultured for 48 h, and observed reduced Ca²⁺ microdomains, similar to those observed at 72 h (see 2-cell stalled embryos shown in image below).

The added RNAseq data is of interest, however an n=2 'Ddx low' embryos is not ideal for analysis. Can the authors confirm whether independent biological samples were used in the validation of the DEGs by qPCR? Further, the authors should detail how GAPDH was selected as the appropriate housekeeper, given its role in energy dynamics and Ca²⁺ signalling. The use of additional housekeepers, or a geometric mean would be more appropriate.

The reviewer is correct. n=2 'Ddx1 low' embryos is not ideal for sequencing analysis. The qPCR data shown in the previous version of the manuscript simply validated the

sequencing data using the same biological samples. In the revised manuscript (extended data Fig. 6), we analyzed another 24 two-cell embryos that were generated by *Ddx1* het/het crosses. Based on the levels of *Ddx1* mRNA, we chose 3 high *Ddx1* and 3 low *DDX1* embryos and used RT-qPCR to validate the sequencing data. These results are presented in the revised extended data Fig. 6

As per the reviewer's suggestion, we also tested two additional housekeeping genes for comparison: *Actb* and *H2az1*. We found that among all three housekeeping genes tested, *Gapdh* has the smallest standard deviation on Ct values within the 24 embryos tested (Ct value for *Gapdh*, *Actb* and *H2az1* was 1.11, 5.33 and 1.52, respectively). For this reason, we selected *Gapdh* for normalization using RT-qPCR (see page 11, lines 226-230).

Have the authors tried isolating ICMs from blastocysts to examine Ca²⁺ localisation using Fluo-Am (given the trouble of uptake by the ICM in intact blastocysts using any molecular probe)?

Unfortunately, we don't have the specialized equipment that would allow us to dissect out the ICM from blastocysts. However, in the previous version of our manuscript, we did use an alternative approach to stain blastocysts (see Figure 5a). For these experiments, we stained the embryos at the morula stage and cultured them for an additional 24 h until they reached the blastocyst stage before imaging. Although the signal was weak, there was no difference between ICM and trophectoderm cells in terms of Ca²⁺ staining pattern. These experiments are described on pg. 11, lines 243-245.

In the conclusion, the authors focus on the potential role of MARVs in modulating mitochondrial function(via Ca²⁺), referencing stalled embryos, however as noted in my previous comments, developmental arrest may equally relate to inadequate activation of the embryonic genome which occurs around the 2-cell stage in mice. This is plausible given the RNAseq data obtained and potential role for MARVs in RNA processing. Based on how embryos have been compared, as noted above, I am not convinced of a cause+effect relationship for MARVs and timed Ca²⁺ release.

We have included the possibility of failed zygotic genome activation in the revised manuscript (pg.12 lines 257-262) which may be related to the regulation of maternal RNA (cytoplasmic polyadenylation or clearance). However, as Reviewer 2 pointed out that we lack hard evidence for this claim, we only briefly allude to this possibility in our revised manuscript.

Please clarify that the mitochondrial number that is halved is 'per cell' (line 92).

Thank you for pointing it out. We now include 'per cell' on line 92.

REVIEWERS' COMMENTS

Reviewer #4 (Remarks to the Author):

The authors have adequately addressed my comments.